# A simple approach to represent precipitation-derived freshwater fluxes into nearshore ocean models: an FVCOM4.1 case study of Quatsino Sound, British Columbia

Krysten Rutherford[1], Laura Bianucci[1], and William Floyd[2,3]

[1]Institute of Ocean Sciences, Fisheries and Oceans Canada, Sidney, British Columbia, Canada
[2]Vancouver Island University, Nanaimo, British Columbia, Canada
[3]British Columbia Ministry of Forests, Nanaimo, British Columbia, Canada

**Correspondence:** Krysten Rutherford (krysten.rutherford@dfo-mpo.gc.ca)

**Abstract.** High-resolution numerical ocean models can be used to help interpret sparse observations in the nearshore as well as to help understand the impacts of climate change and extreme events on these dynamically complex coastal areas. However, these high-resolution ocean models require inputs with comparably high resolution, which is particularly difficult to achieve for freshwater discharge. Here, we explored a simple rain-based hydrological model as inputs into a high-resolution ($\gtrsim 13\,\mathrm{m}$) model of Quatsino Sound - a fjord system located on the northwest coast of Vancouver Island, British Columbia, Canada. Through a series of sensitivity tests using an application of the Finite Volume Community Ocean Model (FVCOM version 4.1), we found that model performance was hindered by the lack of knowledge of ungauged rivers and streams. In this case study, including the only major gauged river implied ignoring 538 other watersheds of various sizes and accounted for only about a quarter of the total estimated freshwater discharge. We found that including at least 60% and ideally closer to 75-80% of total freshwater fluxes gave similar model performance to including all possible 539 freshwater sources; in our model simulations, this percentage of freshwater flux meant including rivers with watersheds greater than $20\text{-}50\,\mathrm{km}^2$, or 7-19 total rivers. Further sensitivity tests also indicated that knowing the main outpour locations into the nearshore ocean is an important factor, but not as important as the total freshwater discharge included. Overall, this study illustrates the complexities of studying the land-ocean connection and offers a simple and accessible tool to help address a common problem in nearshore modelling.

## 1 Introduction

The nearshore ocean is home to rich ecosystems that are dynamically complex and fueled by nutrients from both marine and terrestrial origin, leading to a disproportionate societal and economic importance. Researchers recently defined these dynamic regions at the land-ocean interface as Aquatic Critical Zones (ACZs; Bianchi and Morrison, 2018) - areas that are particularly sensitive to anthropogenic stressors and needing further research. Fjords are one type of ACZ (Bianchi et al., 2020) that are rapidly changing and are notably vulnerable to climate change impacts (e.g. Aksnes et al., 2019; Jackson et al., 2021; Linford et al., 2023). These coastal features were developed by glacial erosion and are characterized as long and narrow inlets with deep, steep-sided channels, often with one or more submarine sills near their mouths (Farmer and Freeland, 1983; Syvitski et al.,

2012). Given their unique topographic features combined with their location directly adjacent to land, fjords are physically and biogeochemically complex regions with processes occurring on relatively short time scales (Syvitski et al., 2012; Bianchi et al., 2020).

Many fjords and other nearshore areas lack enough scientific observations to accurately constrain their highly variable temporal and spatial dynamics. Employing numerical ocean models for coastal areas can thus be an important tool to explore the complex behaviour of these regions; models can help interpret sparse observations to better understand present-day dynamics and anticipate potential future changes. However, the smaller and shorter scales of relevance in the coastal environment require high spatial and temporal resolution to properly represent fjords and other nearshore regions in numerical ocean models. Global Earth System Models (ESMs) have resolution ranging from 10-100 km (Arora et al., 2011; Dunne et al., 2012; Hewitt et al., 2020) and many regional models have resolution ranging from 1-10 km or even greater (e.g. Brennan et al., 2016; Peña et al., 2018; Mortenson et al., 2020; Holdsworth et al., 2021), neither of which are sufficient to accurately constrain the dynamics of nearshore areas. To represent these regions properly, ocean models need resolution less than 100 m (e.g. Foreman et al., 2009; Wu et al., 2014; Deb et al., 2022; Lin and Bianucci, 2023; Foreman et al., 2023).

With increasing model resolution comes the need for high-resolution model inputs, which may at times be difficult to achieve. For instance, freshwater fluxes play an important role in temperate fjord ecosystems, such as those in British Columbia (BC; Farmer and Freeland, 1983; Bianchi et al., 2020). Moreover, small watersheds are the dominant flux of both fresh water and dissolved organic carbon to the nearshore in the northeast Pacific Ocean (McNicol et al., 2023), highlighting the importance of accounting for this source. Accurately constraining freshwater inputs in coastal models is, however, difficult since many rivers and streams are not monitored but still impact these nearshore areas. Such unaccounted for freshwater inputs could decrease model performance and inhibit these models from accurately addressing crucial science questions related to these complex and important nearshore regions.

Previous studies have taken various approaches for estimating freshwater fluxes from rivers and streams without gauges, ranging from simple to more complex. For instance, Royer (1982) used monthly mean precipitation multiplied by a drainage area to estimate freshwater discharge into the coastal waters of the Northeast Pacific. Some nearshore models of BC fjords used watershed area ratios as a way to extrapolate hydrometric data from gauged rivers to nearby ungauged rivers (Foreman et al., 2009; Lin and Bianucci, 2023). Recent studies have improved upon this approach by using watershed characteristics (e.g., glacier or snow cover, reservoirs, etc.) and climate (e.g., mean annual precipitation/snow) to better pair the gauged and ungauged rivers (Foreman et al., 2023; Bianucci et al., 2024). Morrison et al. (2012) and Soontiens et al. (2016) calculated relationships between historical precipitation and hydrometric data to estimate ungauged river flows in BC. On the east coast of Canada, Lambert et al. (2013) and Lavoie et al. (2016) employed a simple hydrological model that uses near-surface air temperature and precipitation-evaporation fields from a regional climate model. Their methods accounted for storage of precipitation as snow when temperatures are below freezing, and factored in a delay in river discharge based on the distance between the runoff grid cells and the mouth of the river. Lastly, Danielson et al. (2020) employed a more complex method, coupling their ocean model of the Gulf of Alaska with a suite of hydrology models that factored in storage of precipitation as snow and in soil, as well as evapotranspiration and routing of runoff across various landscapes (Beamer et al., 2016).

In the present study, we describe a simple hydrological model based on precipitation estimates from a high-resolution atmospheric model, adding to the brief methods described in Royer (1979) and Royer (1982). The simple hydrological model was applied to calculate estimates of river and stream discharge, which were subsequently used as riverine inputs into a high-resolution (13-825 m) Finite Volume Community Ocean Model (FVCOM) application of a fjord system located on the northwest coast of Vancouver Island, BC. While more complex hydrological models are available, the one presented here is easy to set up by oceanographers who may not have access to support from hydrologists. Furthermore, the simplicity of this hydrological model makes it a useful tool for ocean modellers to assess where and what types of more sophisticated hydrological models may be needed for their nearshore models. We evaluated how well this simple hydrological model performed as a source of freshwater inputs into the ocean model and explored various simplifications to the approach to assess how much detail is needed to retain ocean model performance. In particular, we addressed the following two main questions: (1) how much of the total freshwater flux and what subset of the larger watersheds need to be included for optimal model results? and (2) is it important for ocean model performance to know the exact outpour locations of all watersheds? Overall, our study illustrates the importance of better constraining freshwater inputs in nearshore ocean models and presents a simple approach for implementing freshwater fluxes that can be applied to temperate, non-glaciated watersheds.

## 2   Study Region

British Columbia (BC), located on the west coast of Canada, is home to numerous temperate fjords (Pickard, 1961, 1963), making it an ideal region to study the many complexities of fjord ecosystems. The Quatsino Sound fjord system in the west coast of Vancouver Island - used as a case study in the current work - is located on the traditional territory of the Kwakwaka'wakw people and includes Quatsino Sound, Neroutsos Inlet, Holberg Inlet, and Rupert Inlet (Figure 1).

The region has complex bathymetry (Figure 1), with the main channel (Quatsino Sound) and Neroutsos Inlet directly connected to the continental shelf off the west coast of Vancouver Island, and separated from Holberg and Rupert Inlets by a sill at the Quatsino Narrows. The Marble River is this fjord system's largest freshwater source. This river has a watershed area of approximately $512\,\text{km}^2$, is directly connected to Alice Lake and has 2 additional lakes within its watershed, and drains into Rupert Inlet near the mouth of the inlet at its junction with Holberg Inlet and Quatsino Narrows (Drinkwater and Osborn, 1975). It was gauged in the past (1925-1932, and 1969-1971) and more recently since October 2022 (Water Survey of Canada, 2023a). Beyond the Marble River, Quatsino Sound has a diverse population of watersheds that discharge fresh water into the system with rain being a dominant source (Bidlack et al., 2021). Snow contributes to hybrid run-off regimes in higher elevation watersheds that have seasonal snowpack contributing to stream flow in the spring and early summer (Bidlack et al., 2021). Much of the discharge from the region is driven by atmospheric rivers, including the floods on record (Sharma and Déry, 2020). It is expected that as the climate warms, snow will play less of a role in these systems, with rain becoming the main source of discharge (Bidlack et al., 2021).

Historically, this region has seen large impacts from the forestry industry, including erosion, declines in salmon populations, and general changes to the coastal environment (Mustonen et al., 2021). Mine tailings were dumped into Rupert and Holberg

Inlets from the 1970s to 1990s (Pedersen, 1985; Burd, 2002). It is thus clear that large changes have already occurred here since the onset of colonization in the 1700s (Mustonen et al., 2021), and changes will likely continue to affect the region as a result of continued anthropogenic stressors, including increasing climate change impacts.

## 3 Methods

### 3.1 Ocean observations

A comprehensive oceanographic monitoring program was initiated in the Quatsino Sound region in October 2021. Semi-regular (approximately monthly starting in October 2021) Conductivity, Temperature and Depth (CTD) profiles were taken throughout the Sound in the fall of 2021 (see Figure 2 for sampling locations) with the help of the Quatsino First Nation's Fisheries Guardians. A Sea-Bird (SBE-911plus) CTD was used for the 13-15 October 2021 sampling with a sampling interval of 12 Hz; an RBR*concerto* CTD was used for all other CTD profiles in this study (2-5 November and 14-16 December 2021) with an 8 Hz sampling interval.

Two moorings were deployed in Quatsino Sound at 50.43$^o$N, 128.01$^o$W (QUAT1), and 50.48$^o$N, 127.82$^o$W (QUAT2), including the deployment of Acoustic Doppler Current Profilers (ADCPs) and moored CTDs. The CTD and ADCP sensors at QUAT1 were located at 40, 100, and 176 m depth, and at QUAT2 at 40 and 121 m depth. The moored instruments consisted of a Sea-Bird SBE37SMP CTD sensor that took measurements every 10 minutes and Workhorse-ADCPs used in an upwards orientation. The moorings were deployed on October 13, 2021 and retrieved on July 16, 2022.

### 3.2 Ocean model details

For this study, we developed an application of FVCOM (version 4.1; Chen et al., 2003, 2006, 2012) for the Quatsino Sound region. The model domain extends onto the shelf, following the 250 m isobath and encompasses all major inlets (Figure 1): Quatsino Sound, Neroutsos Inlet, Holberg Inlet, and Rupert Inlet. The unstructured, triangular grid has 95,651 nodes and 181,696 elements horizontally, and uses s-coordinates in the vertical with 20 layers (also referred to as general vertical coordinates or tanh sigma coordinate type; upper and lower depth boundary parameters selected as DU=3.0 and DL=0.0, respectively). Horizontal model resolution ranges from 13 to 825 m, with the highest resolution near the coastline in the inlets (horizontal resolution is reported as the square root of the area of each triangle). Wetting and drying was not included, and minimum water depth is set to 10 m as a result. Model bathymetry was taken from the Geological Survey of Canada's *Canada west coast topo-bathymetric digital elevation model* (Kung et al., 2021) and smoothed using 5 iterations of Laplacian smoothing to remove steep bathymetric gradients. Previous tests with the model used a bathymetry smoothed with a volume preserving technique; however, we found that this technique did not sufficiently smooth the model bathymetry and, as a result, the model became overmixed throughout the water column and unstable.

Physical model parameterizations and parameter values were similar to those used in other models of BC fjords, such as those described in Lin and Bianucci (2023), Foreman et al. (2023) and Bianucci et al. (2024). More specifically, the turbulence

model used the Generic Length Scale (GLS) equations and the k-omega scheme (Umlauf and Burchard, 2003). The turbulence stability function was calculated using Canuto et al. (2001), Version A. Background vertical diffusion and viscosity were set to $10^{-5}\,\mathrm{m^2\,s^{-1}}$. Bottom roughness was based on the General Ocean Turbulence Model (Burchard and Bolding, 2001) using a length scale of $10^{-3}$ m and a minimum bottom drag coefficient value of 2.5 x $10^{-3}$. Horizontal diffusivity used the Smagorinksy eddy parameterization (Smagorinsky, 1963) with a coefficient of 0.02. The external timestep was set to 0.075 seconds and ISPLIT parameter was set to 10, making the internal timestep 0.75 seconds.

Initial conditions for salinity (S) and temperature (T) were taken from the regional model Coastal Ice-Ocean Prediction System for the West Coast of Canada (CIOPS-W; Paquin et al., 2021) and merged with CTD observations in the inlets. These observations were sampled on October 14th, 2021 (Figure 2) and excluded from any later analyses. This dataset was interpolated onto a grid and blended with CIOPS-W output for the same day on the shelf. The blending between the gridded observational data and the CIOPS-W model output occurred along a boundary at the mouth of Quatsino Sound using a simple smoothing technique. The model was initialized at 00:00 am on October 14th and run until December 31st, 2021. This simulation period was chosen partly due to the relative abundance of observations over these months. This period is also particularly suitable for our research goals since it overlaps with rainy months in the Pacific Northwest, making it a reasonable period to test our rain-based river proxy, as described below. Model analyses focused on November-December 2021, thereby removing any model spin-up period. Boundary conditions - S, T, sea surface height (SSH) - were also taken from CIOPS-W; S and T were taken from daily output, SSH from hourly output. No nudging was applied at the boundaries. Atmospheric conditions (wind, precipitation, temperature, relative humidity, pressure) were provided by the High Resolution Deterministic Prediction System (HRDPS) atmospheric model with 1 km spatial resolution and a temporal resolution of 30 minutes (MSC Open Data, 2022), which was based on the original release of HRDPS (Milbrandt et al., 2016).

### 3.3 Rivers

#### 3.3.1 Description of local rivers and streams

The Marble River watershed covers almost 25% of the area draining into Quatsino Sound. It drains into the mouth of Holberg and Rupert Inlets near Quatsino Narrows, which is a unique configuration from a typical inlet where large rivers enter at the head of the inlet (Drinkwater and Osborn, 1975). The Marble River's drainage area covers about $512\,\mathrm{km^2}$ and there are three large lakes that serve as additional storage and slow the flux of water from the system. Historical data show peak monthly discharges generally occur in December, with low flows from July through September (Water Survey of Canada, 2023b).

There are an additional 538 watersheds (total area of $1590\,\mathrm{km^2}$) of varied sizes and elevation ranges associated with rivers and streams in the area (Figure 3a). Two of those watersheds are greater than $100\,\mathrm{km^2}$ - one drains into Rupert Inlet at the head of the inlet and the other drains into the main Quatsino Sound near the mouth of the channel (Figure 3b). Four watersheds are between $50\,\mathrm{km^2}$ and $100\,\mathrm{km^2}$ - one drains into Neroutsos Inlet at the head of the inlet, one drains into Holberg Inlet at the head of the inlet, and two drain into the main Quatsino Sound (Figure 3b). Twelve watersheds have areas between $20\,\mathrm{km^2}$ and

$50\,\mathrm{km}^2$ draining into the various inlets (Figure 3c). Most watersheds (499) are less than $5\,\mathrm{km}^2$, with a mean size of $3.8\,\mathrm{km}^2$; the smallest watershed in this study is $0.03\,\mathrm{km}^2$.

While rain is the dominant precipitation input to the watersheds draining into Quatsino Sound, the Marble River and four smaller watersheds (one $\sim 150\,\mathrm{km}^2$, the others smaller than $5\,\mathrm{km}^2$) are climatologically categorized as hybrid snow- and rain-driven runoff regime watersheds (based on 30 years of data, 1981-2010; Bidlack et al., 2021). These five watersheds thus experience both higher flow in fall and early winter from rain/rain-on-snow events and a snow-melt signal in spring (Bidlack et al., 2021). The remaining 534 watersheds are rain-dominated regimes with occasional transient snowpacks that contribute minimally to stream flow. Seasonal snowpacks generally develop above approximately 800 to 1000 m elevation, with shallow transient snowpacks at sea level every few years.

### 3.3.2 River discharge proxy

Due to the lack of gauged rivers and streams, we developed a discharge proxy based on instantaneous precipitation from the HRDPS-1km atmospheric model for each of the 539 watersheds. For each HRDPS grid cell $i$, with a grid cell width $d_i$, and at each time step $t$, there is an instantaneous precipitation given as $r_i$. Assuming that all precipitation immediately becomes river discharge, the instantaneous river flux $R_i$ can be defined for each HRDPS grid cell as:

$$R_i = r_i d_i^2 \tag{1}$$

If we assumed that each watershed had $n$ HRDPS grid cells, then the total instantaneous river flux for each watershed $j$ would be defined as:

$$R_{tot,j} = \sum_{i=0}^{n} r_i d_i^2 \tag{2}$$

However, the actual watersheds may not be properly represented by an integer number of cells. We circumvented this issue by normalizing Equation 2 to account for the actual watershed area, $A_{WS,j}$. This normalization is particularly important since many watersheds are smaller than an HRDPS grid cell of $1\,\mathrm{km}^2$. We used the total area for the $n$ HRDPS grid cells $(A_{HRDPS,j} = \sum_{i}^{n} d_i^2)$ to calculate the normalized instantaneous river flux for each watershed $j$ as:

$$R_{tot,j} = \frac{A_{WS,j}}{A_{HRDPS,j}} \sum_{i=0}^{n} r_i d_i^2 \tag{3}$$

The proxy-calculated river discharge is illustrated in Figure 4, highlighting how much river and stream water might be missing from the model inputs if only Marble River discharge was considered. While the proxy currently only considers precipitation, the method could benefit from using precipitation minus evapotranspiration to enhance its accuracy; however, evapotranspiration in the region was estimated to be minimal during the modelled period compared to precipitation and was

not included in the current work. Evapotranspiration would become more important during the spring and summer in the region. For inputting into the model, river salinity was set to zero and river temperature was set equal to the temperature time series from the nearby Nimpkish River (Water Survey of Canada, 2023a) for all rivers and streams. All river forcing variables (temperature, salinity and proxy-calculated river flux) had a forcing frequency of 30 minutes (e.g., see river discharge time series in Figure 4).

This method does have some drawbacks and limitations, such as assuming that all rain immediately turns into river discharge into the fjord system, which is inaccurate for most watersheds. There will be some storage of the precipitation (e.g. in soil and lakes), which would primarily delay and limit the peak amount of rain that discharges into the fjord system; in the particular case of the Marble River, the lakes within the watershed will partly store some of the precipitation entering this watershed and slow the water as it flows through the system. Furthermore, rainfall in large watersheds would take longer to transit the larger distances, leading to a lag between rain events and peak river discharge into the inlets. The river-proxy method additionally ignores the contribution of ice- and snow-melt to discharge, which would be an issue during warmer periods and when it rains over accumulated snow/ice. In our study region, some snow was present on the ground in late November and December 2021 (Copernicus Sentinel Data, 2023), thus we acknowledge the lack of snow-melt in our model estimates. Nevertheless, this missing contribution is likely overshadowed by the lack of rain storage and transit times, such that the overall result is an overestimation of the instantaneous freshwater flux. Lastly, these methods require assuming river and stream temperature; for example, in the present study, we have assumed that all river and stream discharge temperatures are equal to that measured at the nearby Nimpkish River. The Nimpkish River is a large river attached to a large lake, and its temperature is thus not necessarily representative of many of the rivers/streams draining into Quatsino Sound. However, initial tests have shown that modifications of river temperature have not led to significant changes in fjord density or circulation, likely due to the dominant role of salinity in the stratification of this region.

Despite these limitations, the simplicity and ease of application of the method make it worthy of analysis, as it could provide some information on river and stream outflows that may otherwise be unavailable to and thus ignored by coastal ocean modellers. To illustrate the proxy's ability to estimate river discharge, we evaluated the performance of these methods for the Marble River; the river proxy was applied to the available HRDPS-1km precipitation time series (2021-2022) for comparison against the available discharge data, which has only been collected since October 2022. Although these time periods have minimal overlap ($\sim$ 2 months), the available data can inform the proxy's ability to estimate river discharge. This evaluation is the most difficult test for the proxy, since a method that ignores water storage is not expected to properly represent the discharge from a large watershed with large lakes, like the Marble River (see Appendix C). Nevertheless, the monthly mean river discharge estimated by the river proxy is comparable to the monthly mean discharge measured by the Marble River gauge (Figure 5). Notably, the 2021 river proxy values for September to November were higher than values for the same period in 2022 calculated from both the proxy and from the river gauge. Considering that fall 2021 was a particularly rainy year that included an atmospheric river in mid November, these values are not unreasonable. Furthermore, while the half-hourly proxy dataset shows large spikes, a 7-day running mean suggests that the method produces an appropriate amount of river discharge at weekly timescales (Figure C1). Overall, this comparison illustrated the proxy's ability to reasonably represent the freshwater

discharge at weekly and longer scales, although more complex methods would be more appropriate for large watersheds with large lakes.

## 3.4 Sensitivity tests

A set of sensitivity tests were designed to address the questions outlined at the end of the Introduction (Table 1). The first four sensitivity tests (*Marble River Only, All Rivers, Watersheds50km, Watersheds20km*) were used to identify how much of the total estimated freshwater flux, as measured by the number and size of watersheds included, achieved a balance between model performance and simplicity in the freshwater forcing. The last three sensitivity tests (*Distributed538, Distributed6, Distributed18*) were used to determine the importance of the distribution of total freshwater flux into the Quatsino Sound fjord system. In these three experiments, the Marble River discharge was determined individually using the proxy, while all of the remaining freshwater discharge was aggregated and evenly distributed among different number of watersheds. Throughout the text we refer to pour points, which we define as the outpour locations of freshwater (river, stream) discharge into the ocean model grid.

To focus on the effects of freshwater amounts and distribution, analyses focus on the salinity of the upper 50 m of the water column, where freshwater inputs have the largest impact. To evaluate model performance, modelled salinity values were compared to CTD profile observations from November and December 2021. Model output at the grid node closest to each observation location was selected and linearly interpolated both temporally and vertically to match the observations and create model-observation pairs for evaluation. We calculated the arithmetic mean for most model-observation comparisons, which we will henceforth simply refer to as mean values. Model bias and root mean-square-error were calculated to determine the mean deviation between the modelled and observed salinity values and the deviation in the least-squares, respectively. Full details of these metrics are described in (Lehmann et al., 2009). The nondimensional Willmott skill score (Willmott, 1981) was also calculated; this skill score ranges from zero to one, with one being the best possible score, and indicates the model error divided by the range in values of the observations (Liu et al., 2009). Additional model evaluation (including temperature, tides and current comparisons) is found in Appendix A.

## 4 Results

### 4.1 Model performance

We focused on salinity at the surface and down to 50 m since these regions are the most affected by riverine inputs. Differences in model performance between the *Marble River Only* and the *All Rivers* simulations were first compared to illustrate the two extreme cases presented in Table 1. The different model sensitivity tests are compared to observations from CTD profiles taken throughout the fjord system, as described in Section 3.4.

At the surface, the *Marble River Only* simulation was saltier than the *All Rivers* simulation, with mean surface values up to 4 g/kg greater in some locations (Figure 6a and b). The saltier surface waters resulted from the smaller amount of fresh water

**Table 1.** Description of model simulations.

| Simulation ID | % of total discharge | Description |
|---|---|---|
| *Control simulation* | | |
| Marble River Only | 26.5% | River proxy applied to Marble River; no other rivers/streams included |
| *Simulations with different numbers of watersheds* | | |
| All Rivers | 100% | River proxy applied to all watersheds; all rivers/streams included |
| Watersheds50km | 58.2% | River proxy applied to watersheds $\geq 50$ km$^2$; 7 rivers included |
| Watersheds20km | 76.0% | River proxy applied to watersheds $\geq 20$ km$^2$; 19 rivers included |
| *Simulations with different distributions of freshwater flux* | | |
| Distributed538 | 100% | River proxy applied to Marble River while freshwater flux for all other 538 watersheds is aggregated and evenly distributed amongst the 538 pour points |
| Distributed6 | 100% | River proxy applied to Marble River while freshwater flux for all other 538 watersheds is aggregated and evenly distributed amongst the pour points of the 6 largest watersheds |
| Distributed18 | 100% | River proxy applied to Marble River while freshwater flux for all other 538 watersheds is aggregated and evenly distributed amongst the pour points of the 18 largest watersheds |

into the system relative to the *All Rivers* simulation ($<30\%$ in *Marble River Only* vs. $100\%$ in *All Rivers*; Table 1). Similar differences were noted when looking at the entire top 50 m of the water column. For instance, 2D histograms of salinity from the CTD observations and model at the same time and location showed that although the *Marble River Only* case did a reasonably good job at simulating salinity throughout the region, it overestimated the salinity in the fresher value range (25-30 g/kg, Figure 7a), indicative of the insufficient amount of freshwater discharge entering the model. Conversely, the 2D histogram for the *All Rivers* test (Figure 7b) showed more points on the 1:1 line in the fresher value range (25-30 g/kg), indicating that this model set-up is better able to represent fresher waters.

We further assessed whether the distribution and statistical characteristics of the model were realistic through comparisons of 1D histograms of modelled and observed salinity (Figure 8). Ideally, the modelled salinity histograms would directly overlap the observational salinity histograms; however, the distribution of salinity values in the *Marble River Only* simulation was narrower than the observations, unable to properly capture the fresher values (Figure 8a). In contrast, the *All Rivers* simulation better captured the distribution of salinity, particularly between 25 and 30 g/kg, throughout the modelled region (Figure 8b). All model metrics improved in the All Rivers simulation compared with the *Marble River Only* simulation (Figure 9a, circle and square symbols, respectively): bias of 0.04 vs 0.7 g/kg, RMSE of 1.7 vs 2.2 g/kg, and Willmott Score of 0.86 vs 0.68.

Mean depth profiles of the top 50 m for Quatsino Sound, and Holberg and Rupert Inlets combined (Figure 10) highlight if there were any biases at specific depths in the model results (green) compared to the CTD profiles (yellow). The *Marble River*

*Only* model simulation tended to overestimate salinity throughout the top 50 m by approximately 1 g/kg in Quatsino Sound and by 2.5-3 g/kg in Holberg and Rupert Inlets (Figure 10a and b). For the *All Rivers* case, the mean depth profile of salinity in the top 50 m in the model was much more similar to the observations, with differences between the model and observations of less than 0.5 g/kg in Quatsino Sound and less than 1 g/kg in Holberg and Rupert Inlets (Figure 10c and d).

## 4.2 Determining the optimal fraction of total freshwater flux

Additional sensitivity tests - *Watersheds20km* and *Watersheds50km* (Table 1) - were performed to determine how many of the larger watersheds and what fraction of total estimated freshwater inputs were needed to improve ocean model performance from the *Marble River Only* case. For the case including only the rivers with watersheds greater than $50\,\mathrm{km}^2$, which is 7 rivers total representing 58% of the total discharge, the model performed qualitatively similarly to the *All Rivers* case as a whole. This *Watersheds50km* model simulation fell similarly on the 1:1 line with the CTD observations (Figure 7c) and the model had a comparable salinity distribution (Figure 8c). However, when looking at mean depth profiles, the model was too salty at the surface by 2 g/kg in Holberg-Rupert Inlets and by $\sim 0.5$ g/kg throughout the upper 50 m in Quatsino Sound (Figure 10e,f). In the top 50 m, this model case was biased by 0.3 g/kg, had an RMSE of 1.8 g/kg and a Willmott Score of 0.82 (Figure 9a, inverted triangle symbol).

When we included all rivers with watersheds greater than $20\,\mathrm{km}^2$, a total of 19 rivers representing 76% of the total discharge, the model again performed qualitatively similarly to the *All Rivers* case when looking at the modelled region as a whole. The modelled salinity was distributed similarly to the CTD observations (Figures 7d and 8d). The modelled vertical profiles in this case fell almost exactly on top of the observations (Figure 10g,h) similar to the *All Rivers* case, although it was slightly saltier in the top 50 m. Model metrics were close to those of the *All Rivers* case, with a bias of 0.2 g/kg, a RMSE of 1.7 g/kg and a Willmott Score of 0.85 (Figure 9a, diamond symbol).

## 4.3 Determining the role of freshwater distribution

Through collaboration between oceanographers and hydrologists, the present study included detailed information about the regional watersheds and their associated outpour locations into the Quatsino Sound fjord system. However, this information is not always readily available for ocean modellers. Therefore, additional sensitivity tests (see Table 1) allowed us to assess the impact of the spatial distribution of freshwater discharge (other than the Marble River discharge) on the ocean model performance. As a reminder, the Marble River was not re-distributed since it is the largest river in the region and, as such, usually the only one an ocean modeller would include.

Mean depth profiles of salinity in the top 50 m in Quatsino Sound and Holberg-Rupert Inlets for three sensitivity tests - *Distributed538*, *Distributed6*, and *Distributed18* - showed relatively good performance (Figure 11). The *Distributed538* test performs similarly to the *Watersheds50km* test, with a bias of 0.3 g/kg, an RMSE of 1.7 g/kg and Willmott Score of 0.8 (Figure 9a, triangle symbol). That said, this simulation has one of the largest salinity biases and RMSE values in both Holberg and Rupert Inlets (Figure 9c,d), which was particularly noticeable in the upper 5 m, where it is too salty by about 1.5-2 g/kg (Figure 11a). Both the *Distributed6* and *Distributed18* performed similarly to the *All Rivers* and *Watersheds20km*

sensitivity tests. *Distributed6* has a bias of 0.1 g/kg, an RMSE of 1.6 g/kg and Willmott Score of 0.87, and *Distributed18* has a bias of -0.01 g/kg, an RMSE of 1.6 g/kg and a Willmott Score of 0.89 (Figure 9a, star and cross symbols, respectively). The *Distributed6* and *Distributed18* simulations performed similarly in Quatsino Sound, and Rupert and Neroutsos Inlets (Figure 9b,d,e); *Distributed18* improved over *Distributed6* in Holberg Inlet (Figure 9c).

## 5  Discussion

The overarching goal of this study was to use a simple approach for estimating river discharge as a tool to highlight how nearshore ocean models can benefit from improved understanding of freshwater fluxes. The approach applied here used a high-resolution (1 km) precipitation product multiplied by watershed areas to estimate freshwater discharge going into the Quatsino Sound fjord system, reminiscent of the river discharge estimates calculated from monthly mean precipitation in Royer (1979) and Royer (1982). We used the ocean model to identify the amount of freshwater discharge as well as how many and what size of watersheds are required in the model simulations for optimal results. We additionally explored the importance of the spatial distribution of the total freshwater flux.

The *Marble River Only* model simulation performed the worst of all sensitivity tests throughout the entire model domain (Figure 9a). This experiment was on average saltier, by upwards of 4 g/kg in some areas, than the other sensitivity tests that include more rivers and streams (Figure 6). The *Marble River Only* case was unable to capture the fresh values in the observations (Figures 7, 8) and was consistently too salty throughout the upper 40 m of the water column (Figure 10). Furthermore, this *Marble River Only* simulation had the worst model metrics, with a salinity bias of 0.7 g/kg, RMSE of 2.1 g/kg and Willmott Score of 0.68 in the top 50 m (Figure 9a). The *All Rivers* simulation performed considerably better than the *Marble River Only* simulation, with a salinity bias of 0.04 g/kg, RMSE of 1.7 g/kg and Willmott Score of 0.87 in the top 50 m. All other sensitivity tests had metrics with salinity bias less than 0.4 g/kg, RMSE less than 2 g/kg and Willmott Score higher than 0.8. Overall, these saline characteristics of the *Marble River Only* test indicated that this simulation does not have enough fresh water reaching the fjord system.

The large RMSE values in the *All Rivers* simulation were largely due to discrepancies in Neroutsos Inlet as well as Holberg Inlet - RMSE values in both Quatsino Sound and Rupert Inlet were $\sim 0.5$ vs 1.9 g/kg in Neroutsos Inlet and 2.8 g/kg in Holberg Inlet (Figure 9b-e). Previous work has shown that Holberg Inlet is a relatively unique inlet compared to other BC inlets, such as those on the mainland (Pickard, 1961, 1963; Drinkwater and Osborn, 1975). Drinkwater and Osborn (1975) believed that strong tidal mixing through Quatsino Narrows in combination with the large freshwater discharge from the Marble River at the mouth of Holberg Inlet make this part of the system particularly unique. These features create steep vertical gradients in salinity (Figures 10, 11) that can be difficult to simulate. Less work has been done to study mixing in Neroutsos Inlet, however it is a very deep and steep-sided channel (Figure 1) and all sensitivity tests struggled to substantially improve the model's RMSE here (Figure 9e). It is likely that the complex bathymetry in Neroutsos Inlet creates unique mixing and steep vertical gradients that are again difficult to simulate. As noted in Foreman et al. (2023), horizontal diffusion in FVCOM occurs parallel to the terrain-following vertical layers (Chen et al., 2006), which can lead to overly diffusive thermoclines and haloclines.

Therefore, approximations within FVCOM could explain both the model's inability to capture the steepness of the halocline in Holberg Inlet as well as the overly salty waters at the surface in some locations in the model (Figure 10). Future work will aim to better understand model limitations in this regard and more comprehensively evaluate mixing within the Quatsino Sound fjord system, including fine-tuning bathymetry (i.e. minimize the current bathymetric smoothing while retaining model performance) and testing the wetting-drying function.

Although watershed analyses established that 539 rivers and streams flow into the Quatsino Sound fjord system, our results indicate that good model performance can be achieved as long as 60% or more of the fresh water is accounted for. Compared with the Marble River Only simulation, model performance improved just by including rivers with watersheds greater than 50km$^2$ (7 rivers total and ~58% of total flux; Figures 7-10) and including rivers with watersheds greater than $20\,\text{km}^2$ improved the model results even further (19 rivers total and ~ 75% of total flux; Figures 7-10). The latter sensitivity test performed very similarly to the *All Rivers* simulation (Figures 6 - 10); however, it is worth noting that the *Watersheds20km* simulation had a larger salinity bias than the *All Rivers* case in the upper 50 m in both Holberg (0.5 vs. 0.2 g/kg) and Rupert (0.25 vs. <0.1 g/kg) Inlets (Figure 9). We conclude that including only the major rivers is not enough to achieve good coastal model performance (e.g., our Marble River Only simulation). Our results also suggest that if watershed information is limited, even including a fraction of the total freshwater sources will improve ocean model performance; in our case study, including at least 60-75% of total freshwater sources substantially improved upon the Marble River Only simulation. While the exact amount of freshwater discharge required will depend on the specific system being modelled, these results might be useful for guiding model development for other regions.

If the locations of the fresh water pour points are not known except for the major rivers, our results showed that improved model performance is still seen when distributing the total watershed flux throughout the domain (Figure 9, 11). However, the sensitivity tests with the total flux distributed over the largest watersheds' pour points performed better than the case with the total riverine flux evenly distributed throughout the domain (Figure 11). These results indicate that the ocean model benefited from some knowledge of the local watersheds since including all freshwater inflow in just the larger watersheds improves the model performance. However, including the total amount of fresh water remains a more crucial step, as seen by the contrast between the *Marble River Only* or *Watersheds50km* and *Distributed538* simulations (i.e., the latter improved metrics and overall performance over the other two, Figures 9 - 11).

Despite the large improvements in ocean model performance by applying this rain-based river proxy to the regional watersheds, the proxy itself has drawbacks and limitations. This approach assumed that all rain immediately becomes river discharge, which ignores any lag between the peak rainfall and peak discharge due to retention, storage, and transit times. For example, Drinkwater and Osborn (1975) estimated that there was a 1-2 day lag between peak precipitation and peak discharge from the Marble River. Other studies attempted to account for this lag; for example the simple hydrological model presented in Lambert et al. (2013) factors in a delay between precipitation event and river discharge by considering the distance between the runoff grid cell and the river mouth. Aside from the timing of the peak discharge, precipitation will be stored for periods of time in the ground and in lakes, which may affect the total amount and timing of river discharge reaching the nearshore environment.

These assumptions combined result in somewhat unrealistically large instantaneous discharge values for some watersheds, most notably for the Marble River where discharge values reach as high as $\sim 1000\,\mathrm{m}^3\mathrm{s}^{-1}$ (Figure 4).

Furthermore, this method was developed to represent rain-dominated watersheds and is not applicable to snow- and glacier-dominated watersheds. Other studies have used parameterizations (Cowton et al., 2015) and somewhat complex hydrological models (Liston and Mernild, 2012; Danielson et al., 2020) to account for glacial and snow-dominated watershed influence on nearshore ocean models. Even in rain-dominated watersheds, there might be storage of precipitation as snow during the winter months, which can vary from year to year - this process is not accounted for in our methods. Other studies (e.g. Royer, 1982;

Lambert et al., 2013) use air-temperature to determine whether precipitation is stored as snow. For example, $0^o$C can be used as a cutoff: when air temperature is below this threshold, precipitation is stored as snow, and in the spring, when air temperature rises above $0^o$C, snow is gradually released as river runoff. These methods could be similarly added to the simple rain-based hydrological model presented here.

     Despite these drawbacks, the simplicity of the method presented here combined with the noticeable improvements in model

performance make this approach useful for many nearshore modellers. The only requirements to estimate river and stream runoff through this approach are (1) a rudimentary knowledge of watershed area and, ideally, outpour locations, and (2) precipitation (or precipitation minus evapotranspiration or evaporation) from an atmospheric model. If these data are available, these methods are achievable for many ocean modelling projects and could improve the representation of the land-ocean connection in regions where rain is a major driver of local hydrology, as shown here. Furthermore, and in spite of its limitations, the river

proxy method can be applied to identify the watersheds with the largest impact on nearshore ocean models, and that should either be gauged or targeted with a more sophisticated hydrological model. Future work will further evaluate the proxy as more Marble River gauge data becomes available and compare the proxy's performance as a source of fresh water into ocean models against a more complex rainfall-runoff model. Improvements to the proxy could be considered, such as accounting for storage of precipitation as snow cover and the effects of snow-melt for the spring-freshet.

This study underscores the importance of interdisciplinary teams; fostering collaborations between hydrologists and oceanographers is crucial to improving our representations of freshwater fluxes into nearshore ocean models and enhancing our understanding of the land-ocean interface. With climate change impacts only worsening over the coming decades, it is important to ensure our coastal models are able to simulate the dynamic conditions at ocean margins. Climate change is expected to cause a northward shift in storm tracks (Yin, 2005), which will create increasing frequency and severity of extreme weather in

the Northeast Pacific, including atmospheric river events (e.g. Salathé Jr, 2006; Radić et al., 2015). Projections from several climate models suggest that for both RCP4.5 and RCP8.5, conditions for 2070-2100 show an increased frequency of extreme atmospheric river events in BC, and an overall average increase in total precipitation in BC for the autumn season (mid-August to December; Radić et al., 2015). There will also be a shift away from snow contributing to stream flow, especially on Vancouver Island (Bidlack et al., 2021). With such increases in precipitation expected, simple but effective methods for describing

freshwater influence on the nearshore ocean will be important to ensure we can simulate future conditions and anticipate future nearshore changes.

## 6 Conclusions

The simple rain-based hydrological model applied in the present study effectively estimated river discharge into the Quatsino Sound fjord system and demonstrated the need to better constrain river fluxes into nearshore ocean areas. Through a series of sensitivity tests, we showed that including rivers with watersheds larger than 20-50 km$^2$ and $\sim$75% of total freshwater fluxes into nearshore areas best balances model performance with simplicity of implementation. If information on the outpour location of all rivers and streams is not available to ocean modellers, some knowledge of where the largest discharges occur gives better results than evenly distributing the total freshwater flux across the domain. Future work will apply a simple rainfall-runoff model to better account for routing and storage of fresh water in the watersheds draining into Quatsino Sound (e.g. Mockler et al., 2016). Overall, the methods described here can be applied to other nearshore ocean models that are connected to temperate, rain-dominated watersheds. We propose that this approach is an accessible and effective way to improve our understanding of the dynamic areas at the land-ocean interface.

## Appendix A: Additional ocean model evaluation

Additional model evaluation is summarized below. Observations are compared against the *All Rivers* simulation (Table 1).

### A1 Tides

Tidal evaluation of the model was performed by comparing the modelled sea surface height to tidal gauge data located at 5 locations throughout the fjord system (see Figure A1). A comparison of the model (black) against the Bergh Cove tidal gauge observational data (red) is shown (Figure A2) which is representative of the model's performance at the other 4 tidal gauges. Overall the model captures the amplitude of the main tidal signal well without a phase lag.

More detailed comparisons of the phase and amplitude of 5 tidal constituents (M2, K1, S2, O2, N2) at all 5 tidal gauge stations further illustrate the model's ability to simulate tides in the Quatsino Sound fjord system (Table A1). Complex distances up to $\sim$5% of the observed amplitude indicate good model-observation agreement (Foreman et al., 2012). The two main constituents, M2 and K1, are properly represented at all stations except for M2 at Port Alice (PA; complex distance is 12.9 cm or 12% of the observed amplitude). The PA tidal gauge station is located in Neroutsos Inlet, which is the region the model struggles the most to simulate due to its complex bathymetry; other constituents are also poorly represented at this location (e.g. S2, O2). Performance at Makwazniht Islet (MI) is also poor for S2 and O2 constituents, likely due to the complexity of the tides and overall circulation at this location near Quatsino Narrows.

### A2 CTD Profiles: Temperature comparisons

Evaluation of modelled temperature against CTD profiles in the top 50 m of the water column (Figure A3) showed that the distribution of temperature was reasonably similar between the model and the observations. The model overall tends to under-estimate temperature and does not capture some of the warmer values in the domain. In the top 50 m, the model is biased by

**Table A1.** Tidal constituent comparison between the *All Rivers* model simulation and the tidal gauge observational data at 5 different stations (Figure A1).

| Tidal constituent | Amplitude (m) | | Phase ($^o$) | | Complex |
|---|---|---|---|---|---|
| | *Observation* | *Model* | *Observation* | *Model* | distance (cm) |
| **Winter Harbour Tidal Gauge** | | | | | |
| M2 | 1.01 | 1.04 | 358.6 | 2.4 | 7.6 |
| K1 | 0.46 | 0.47 | 244.1 | 244.9 | 1.3 |
| S2 | 0.29 | 0.28 | 257.6 | 265.4 | 4.0 |
| O2 | 0.26 | 0.26 | 336.4 | 334.6 | 0.8 |
| N2 | 0.24 | 0.22 | 195.3 | 194.5 | 2.0 |
| **Port Alice Tidal Gauge** | | | | | |
| M2 | 1.05 | 0.93 | 359.7 | 1.8 | 12.9 |
| K1 | 0.46 | 0.46 | 244.4 | 244.8 | 0.9 |
| S2 | 0.30 | 0.25 | 259.0 | 334.9 | 34.1 |
| O2 | 0.26 | 0.25 | 336.7 | 264.9 | 29.7 |
| N2 | 0.25 | 0.20 | 196.5 | 193.5 | 5.2 |
| **Bergh Cove Tidal Gauge** | | | | | |
| M2 | 1.04 | 1.05 | 360.0 | 3.7 | 6.7 |
| K1 | 0.46 | 0.47 | 244.7 | 245.5 | 0.8 |
| S2 | 0.30 | 0.28 | 259.3 | 266.8 | 4.2 |
| O2 | 0.26 | 0.26 | 337.1 | 335.4 | 0.8 |
| N2 | 0.25 | 0.22 | 196.6 | 195.4 | 2.5 |
| **Coal Harbour Tidal Gauge** | | | | | |
| M2 | 1.04 | 1.05 | 19.8 | 20.8 | 2.1 |
| K1 | 0.47 | 0.48 | 257.9 | 257.0 | 1.8 |
| S2 | 0.28 | 0.27 | 285.4 | 289.5 | 2.4 |
| O2 | 0.25 | 0.27 | 350.2 | 346.3 | 2.3 |
| N2 | 0.24 | 0.22 | 220.1 | 216.8 | 2.8 |
| **Makwazniht Islet Tidal Gauge** | | | | | |
| M2 | 1.04 | 1.04 | 20.0 | 19.9 | 0.2 |
| K1 | 0.45 | 0.49 | 254.4 | 255.7 | 4.0 |
| S2 | 0.30 | 0.27 | 283.0 | 344.5 | 29.4 |
| O2 | 0.25 | 0.27 | 349.2 | 288.2 | 26.4 |
| N2 | 0.24 | 0.22 | 224.0 | 215.9 | 3.7 |

-0.26$^o$C, has an RMSE of 0.71$^o$C and Willmott Score of 0.83. When looking at the individual inlets, we find that the model: is biased by -0.23$^o$C, has an RMSE of 0.38$^o$C and Willmott Score of 0.93 in Quatsino Sound; is biased by 0.26$^o$C, has an RMSE of 0.92$^o$C and Willmott Score of 0.85 in Holberg Inlet; is biased by -0.70$^o$C, has an RMSE of 0.89$^o$C and Willmott Score of 0.68 in Neroutsos Inlet; and is biased by 0.17$^o$C, has an RMSE of 0.33$^o$C and Willmott Score of 0.94 in Rupert Inlet. Overall the model does a reasonable job at representing temperature throughout the model domain, as illustrated by the distributions presented in Figure A3 and the model metrics; more work will be done to further improve the model's representation of temperature. In particular, efforts must be directed at improving the riverine temperatures, which are currently oversimplified (i.e., using the nearest river temperature gauge at the Nimpkish River for all freshwater fluxes into the ocean model).

## A3    Moored CTDs: Temperature and salinity

Comparisons of the model against the moored CTDs at 40 m depth showed that the model does a good job at representing salinity (Figure A4d,e) and a reasonable job at representing temperature (Figure A4b,c) at these locations and depth. More specifically, the temporal variability of salinity at 40 m depth is well captured, although there are some periods where the model is biased slightly too salty compared to the observations, most notably at QUAT2 (Figure A4d). In terms of temperature, the model tends to be slightly cold-biased compared to the observations and it misses some of the short-term (hourly, daily) temporal variability. However, the overall temperature trend over this period is well captured (Figure A4b,c).

## A4    Currents

Comparisons of de-tided current velocities between the model and ADCP observations in the upper 50 m illustrate a good agreement (Figures A5 - A6). Qualitatively, there is agreement in terms of both magnitude and direction between the model and ADCP observations. The variability is well captured, although the model does not always capture the timing of some events. For example, at QUAT1, the overall magnitude of the velocity is well represented in both the eastward and northward velocities, but specific events (e.g. between November 20th to 30th in eastward velocities) and some temporal variability are not present in the model. Conversely, some events are very well captured in the model (e.g. the magnitude and direction of northward velocities at QUAT1 around December 1st). The eastward velocity at QUAT2, which includes some of the highest velocities at these stations, is particularly well captured in the model; the direction and variability of the northward velocities at QUAT2, although much weaker, are also well captured. The timing of events seems to be mostly well represented in the model at QUAT2, see for example comparisons around November 1st or December 5th in both the eastward and northward velocities.

## A5    CTD profiles: Full water column evaluation

Additional evaluation of modelled salinity and temperature against CTD profiles spanning the full water column showed reasonable agreement (Figure A7). The distribution of modelled salinity agrees well with the CTD observations (Figure A7a), particularly if we exclude Neroutsos Inlet from the analysis (Figure A7b). As discussed, the model shows over-mixed condi-

**Table B1.** Model metrics shown in Figure 9 for seven sensitivity tests: All Rivers, Marble River Only (MRivO), Watersheds50km (W50km), Watersheds20km (W20km), Distributed538 (D538), Distributed6 (D6) and Distributed18 (D18).

| Region | All Rivers | MRivO | W50km | W20km | D538 | D6 | D18 |
|---|---|---|---|---|---|---|---|
| *Root mean square error (RMSE, g/kg)* | | | | | | | |
| Full domain | 1.69 | 2.16 | 1.81 | 1.72 | 1.85 | 1.63 | 1.58 |
| Quatsino Sound | 0.46 | 0.76 | 0.48 | 0.44 | 0.42 | 0.50 | 0.48 |
| Holberg Inlet | 2.16 | 3.53 | 3.13 | 2.92 | 3.25 | 2.76 | 2.58 |
| Rupert Inlet | 0.48 | 1.13 | 0.73 | 0.56 | 0.61 | 0.44 | 0.45 |
| Neroutsos Inlet | 1.87 | 2.33 | 1.88 | 1.87 | 1.93 | 1.76 | 1.77 |
| *Bias (g/kg)* | | | | | | | |
| Full domain | 0.04 | 0.71 | 0.34 | 0.17 | 0.27 | 0.10 | -0.01 |
| Quatsino Sound | 0.09 | 0.40 | 0.23 | 0.15 | 0.15 | 0.09 | 0.04 |
| Holberg Inlet | 0.15 | 1.25 | 0.80 | 0.46 | 0.56 | 0.28 | -0.11 |
| Rupert Inlet | 0.08 | 0.83 | 0.49 | 0.26 | 0.24 | 0.11 | -0.10 |
| Neroutsos Inlet | -0.07 | 0.70 | 0.16 | 0.02 | 0.23 | 0.00 | 0.03 |
| *Willmott Score* | | | | | | | |
| Full domain | 0.87 | 0.68 | 0.82 | 0.85 | 0.80 | 0.87 | 0.89 |
| Quatsino Sound | 0.96 | 0.83 | 0.95 | 0.96 | 0.96 | 0.96 | 0.96 |
| Holberg Inlet | 0.82 | 0.64 | 0.74 | 0.79 | 0.70 | 0.83 | 0.86 |
| Rupert Inlet | 0.95 | 0.65 | 0.85 | 0.92 | 0.90 | 0.95 | 0.96 |
| Neroutsos Inlet | 0.82 | 0.40 | 0.79 | 0.81 | 0.68 | 0.82 | 0.79 |

tions in this deep and narrow inlet (see more in Section 5). Modelled temperature adequately compares to CTD temperature observations throughout the entire water column (Figure A7c), with a similar comparison as in the top 50 m (Figure A3a)). More specifically, the model has a smaller temperature range than the observations, unable to capture the cooler ($<8^{o}$C) and warmer ($>12^{o}$C) temperatures, in large part due to the issues encountered in Neroutsos Inlet (Figure A7d). Future work will perform a more rigorous evaluation of the full water column.

**Appendix B: Values of model metrics in Figure 9**

To supplement Figure 9, here we display model metrics for all sensitivity tests in a different format (Table B1) for more clarity on each model simulation's performance.

## Appendix C: Additional evaluation of the rain-based river proxy

To further illustrate the ability of the rain-based proxy to provide reasonable amounts of fresh water to the Quatsino Sound fjord system, we show comparisons of the proxy (applied to the Marble River watershed), its 7-day running mean, and the Marble River gauge data for October and November 2022 (Figure C1). The proxy and gauge data only overlap for this short period at the end of 2022. While the proxy time series (grey; data points every 30 minutes) shows large spikes due to its assumptions and limitations (see Section 3.3.2), the 7-day running mean of the proxy (black) produces more realistic values in alignment with the data from the Marble River gauge (light green). While the magnitude of the smoothed proxy is similar to the Marble River gauge data, the limitations of the methodology (e.g., no storage of rain in soil or lakes) are still evident. For instance, the 7-day running mean of the proxy produces river discharge that peaks earlier than the gauge data during rain events (e.g. November 20-30th). Additionally, the smoothed proxy time series reaches a minimum value of 0 m$^3$ s$^{-1}$ in between rain events whereas the gauge data does not (see around November 15th). These differences are largely due to the presence of large lakes within the Marble River's watershed, as well as its relatively large size. The proxy thus overestimates the Marble River's discharge on shorter timescales and is unable to capture the timing of peak flow. Despite these differences, the proxy is able to deliver adequate amounts of fresh water to the system over a 7-day period and is expected to perform better in the smaller watersheds that do not have lakes. More complex methods, such as sophisticated rainfall-runoff models or incorporating river gauge data when available, would ideally be employed for watersheds that cover a larger area and/or include large lakes.

*Code and data availability.* FVCOM code is developed by the Marine Ecosystem Dynamics Modeling Laboratory (MEDM-Lab) at the University of Massachusetts and is publicly available by their developers at https://github.com/FVCOM-GitHub/FVCOM under the MIT/X license; a fixed version of the code and input files used in this case study can be found at https://doi.org/10.5281/zenodo.10602542 (Rutherford et al., 2024a). Observational and model datasets used in this manuscript are available at https://doi.org/10.5281/zenodo.10602511 (Rutherford et al., 2024b).

*Author contributions.* KR led model development, performed model simulations, produced figures, and led the writing of the manuscript. LB conceptualized the project, obtained funding, and regularly discussed results and writing. WF contributed to developing the rain proxy. All coauthors contributed to editing the manuscript.

*Competing interests.* The authors declare that they have no competing interests.

*Acknowledgements.* We want to acknowledge that this work studies a region of the traditional territory of the Quatsino First Nation (QFN), who we are thankful to work with on the observational aspects of this project. In particular, we appreciate the help from the QFN Fisheries

Guardians, Graem Hall and Mark Wallas, and the Fisheries Manager, Kira Sawatzky. We thank Glenn Cooper and Colin Webber for helping collect and process some of the observational datasets, Andy Lin for his help with the initial model development and advice along the way, and Warren Olmstead for his help delineating watershed areas. From Mowi West, we want to thank Alexandra Eaves, Kevin Kowalchuk, and Kelly Osborne for their input and help organizing aspects of this project. We are grateful for the advice and discussions from members of the UBC-DFO Modelling Working Group, in particular invaluable feedback from Susan Allen. Jonathan Izett provided helpful feedback during

an internal DFO review before submission. This work was funded by Fisheries and Oceans Canada's Aquaculture Collaborative Research Development Program (ACRDP).

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

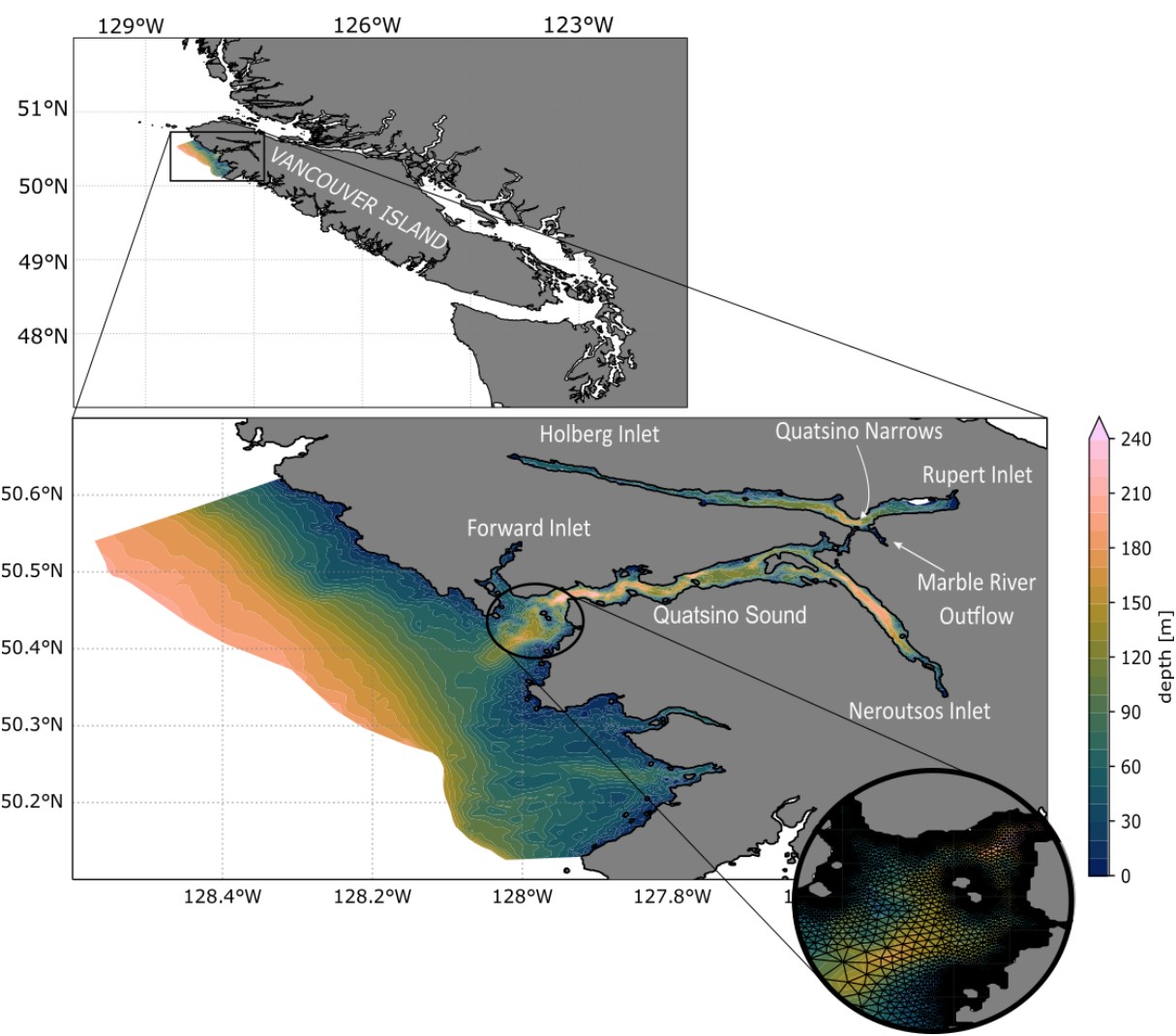

**Figure 1.** Top panel shows a map of Vancouver Island with the location of the study region, Quatsino Sound, bounded by the black rectangle. Bottom panel zooms in on the model domain and Quatsino Sound fjord system. The colour-scale indicates model depth and the main features in the region are indicated with white text. The circular inset shows an example of the model grid, which is triangular and ranges in resolution from 13-825 m.

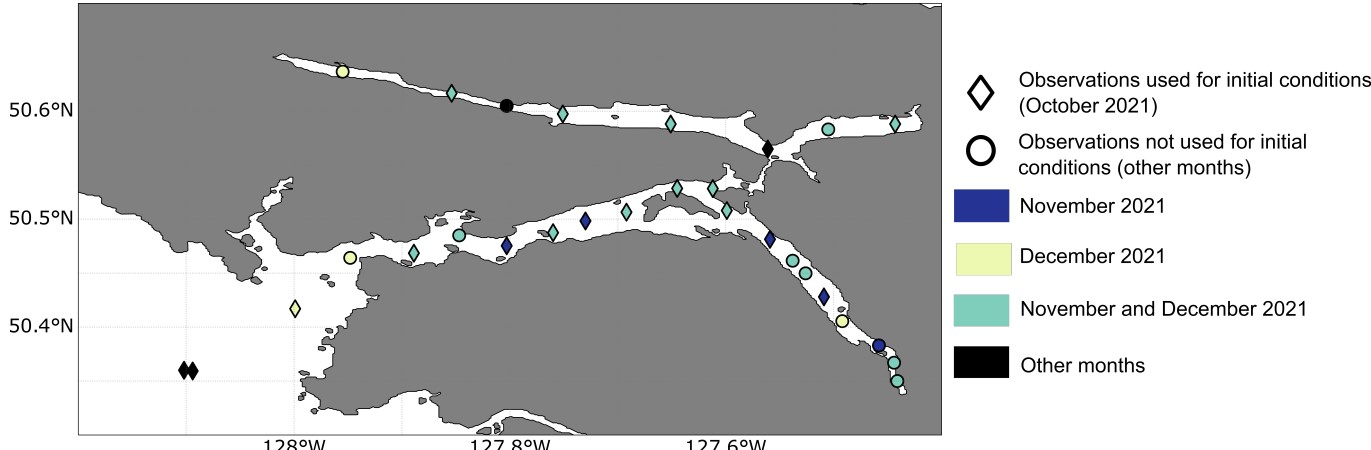

**Figure 2.** Summary of CTD sampling locations. Diamonds are locations where observations were taken between October 13-15, 2021 and used to create temperature and salinity initial conditions for the model simulations. Circles indicate locations where observations were taken on other months. Colouring further describes the timing of observations: November 2021 (dark blue), December 2021 (light yellow-green), both November and December (light blue-green), and other periods (black).

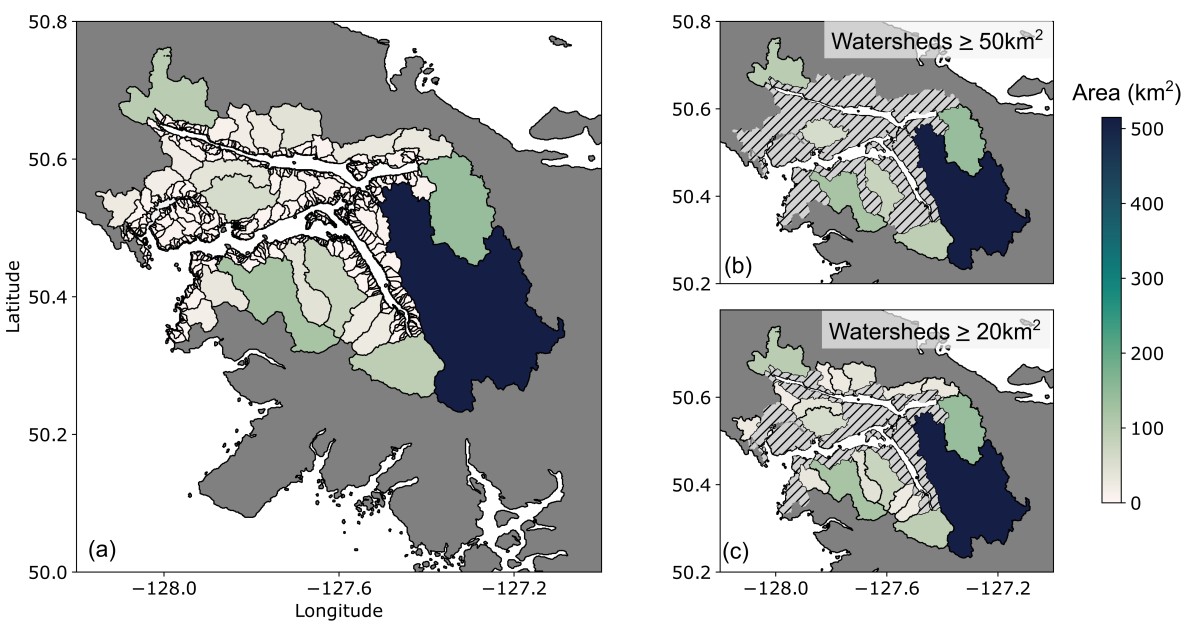

**Figure 3.** Watersheds that drain into the Quatsino Sound fjord system. Colour indicates watershed area. The largest watershed (Marble River) is $512\,\text{km}^2$ and highlighted in dark blue. (a) Shows all watersheds, (b) shows only the watersheds that have an area greater than or equal to $50\,\text{km}^2$ (7 watersheds total), and (c) shows only the watersheds with an area greater than or equal to $20\,\text{km}^2$ (19 watersheds total).

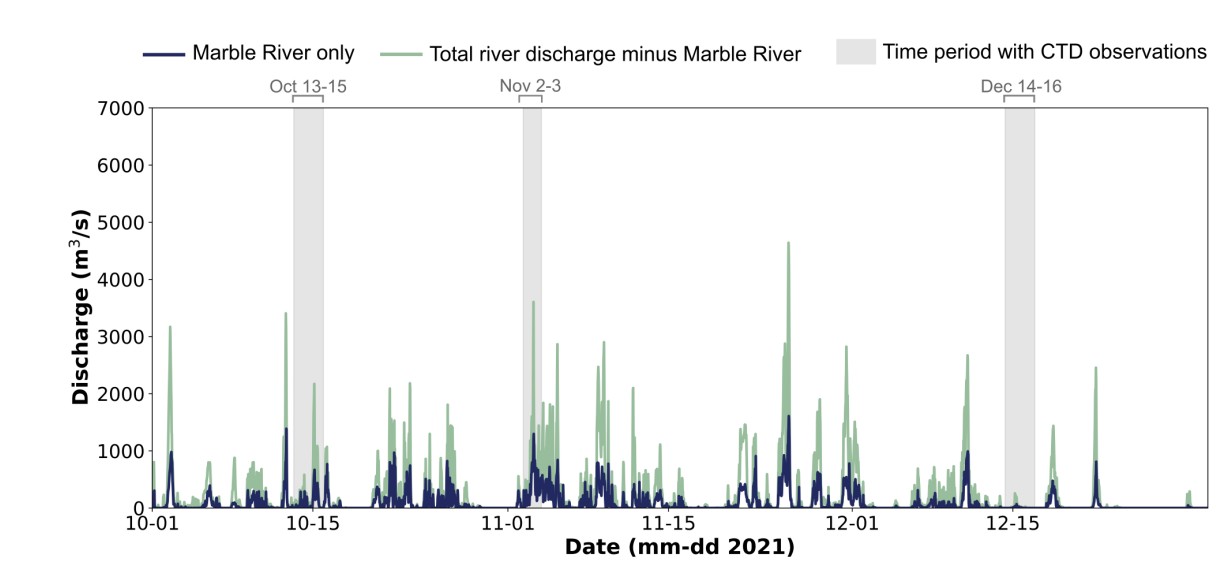

**Figure 4.** Time series of instantaneous river discharge ($m^3$ $s^{-1}$) calculated from the precipitation proxy outlined in equation 3. The dark blue line indicates the river discharge from the Marble River and the light blue-green line indicates total river discharge from all other watersheds draining into the Quatsino Sound fjord system. Watershed areas are indicated in Figure 3. The grey shaded areas indicate when CTD observations were taken (see Figure 2 for CTD sampling locations).

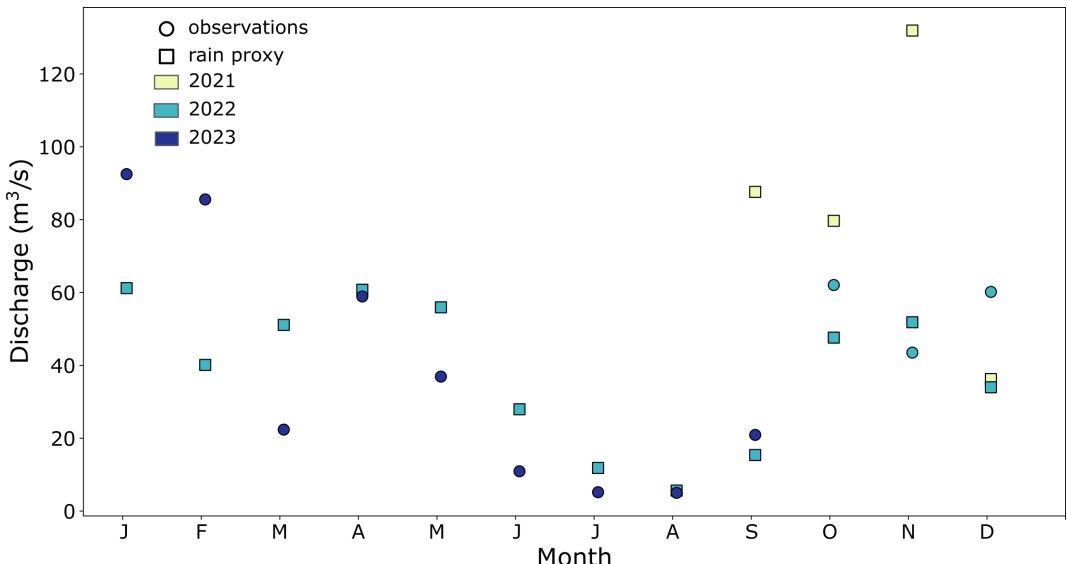

**Figure 5.** Comparisons of monthly mean river discharge calculated using the rain proxy (square) versus Marble River discharge data (circle) from the Water Survey of Canada (Water Survey of Canada, 2023a). Marble River discharge data are available starting in October 2022, which does not overlap with the simulation period. The rain proxy is calculated from September 2021 to December 2022 due to HRDPS-1km availability.

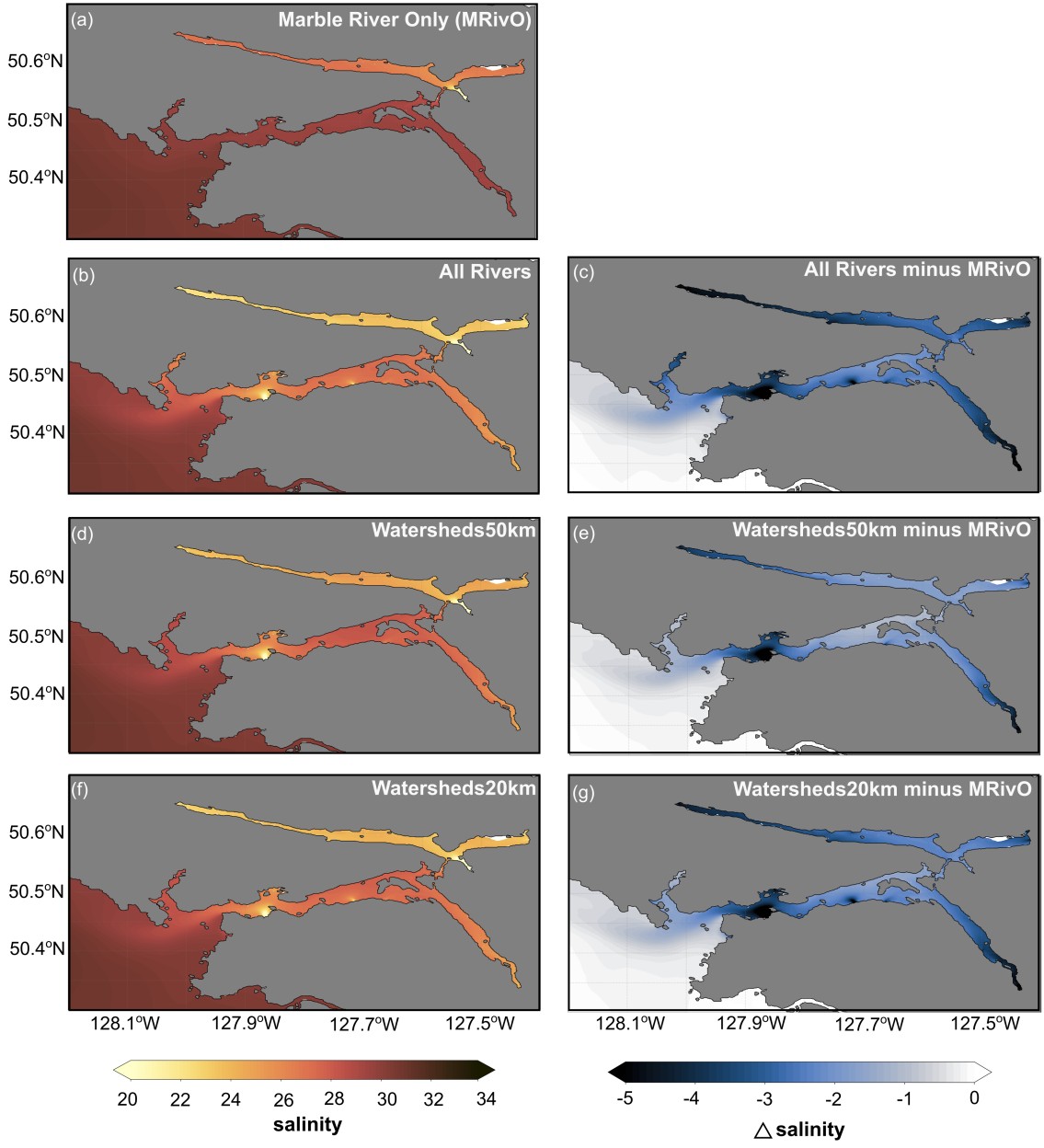

**Figure 6.** Mean surface salinity (left panel) and difference in mean surface salinity between the *Marble River Only* (MRivO) case and subsequent sensitivity tests (right panel) in November 2021 for (a) Marble River Only, (b,c) All Rivers, (d,e) Watersheds50km, and (f,g) Watersheds20km.

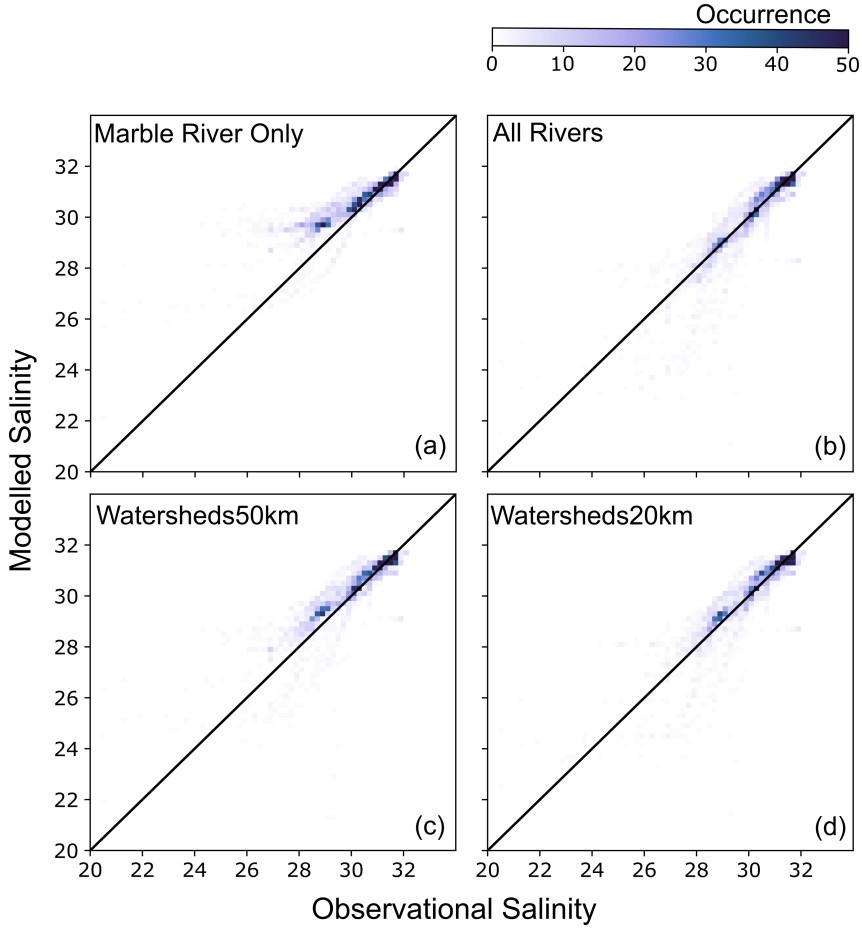

**Figure 7.** 2D histograms of model performance in the top 50 m for (a) Marble River Only, (b) All Rivers, (c) Watersheds50km, and (d) Watersheds20km. The black line indicates the 1:1 line, which is ideally where the highest occurrence (darker colours) would fall. Values above the line indicate that the model is too salty and values below the line indicate that the model is too fresh.

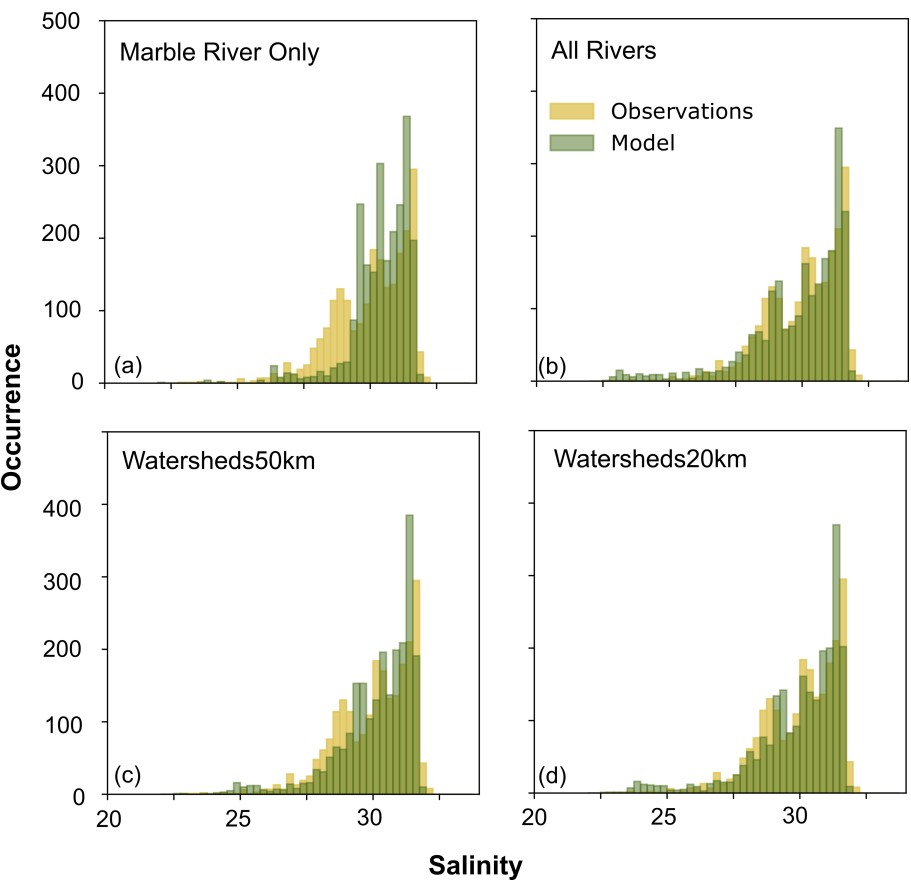

**Figure 8.** Histograms of salinity for the model (green) vs observations (yellow) in the top 50 m for (a) Marble River Only, (b) All Rivers, (c) Watersheds50km, and (d) Watersheds20km.

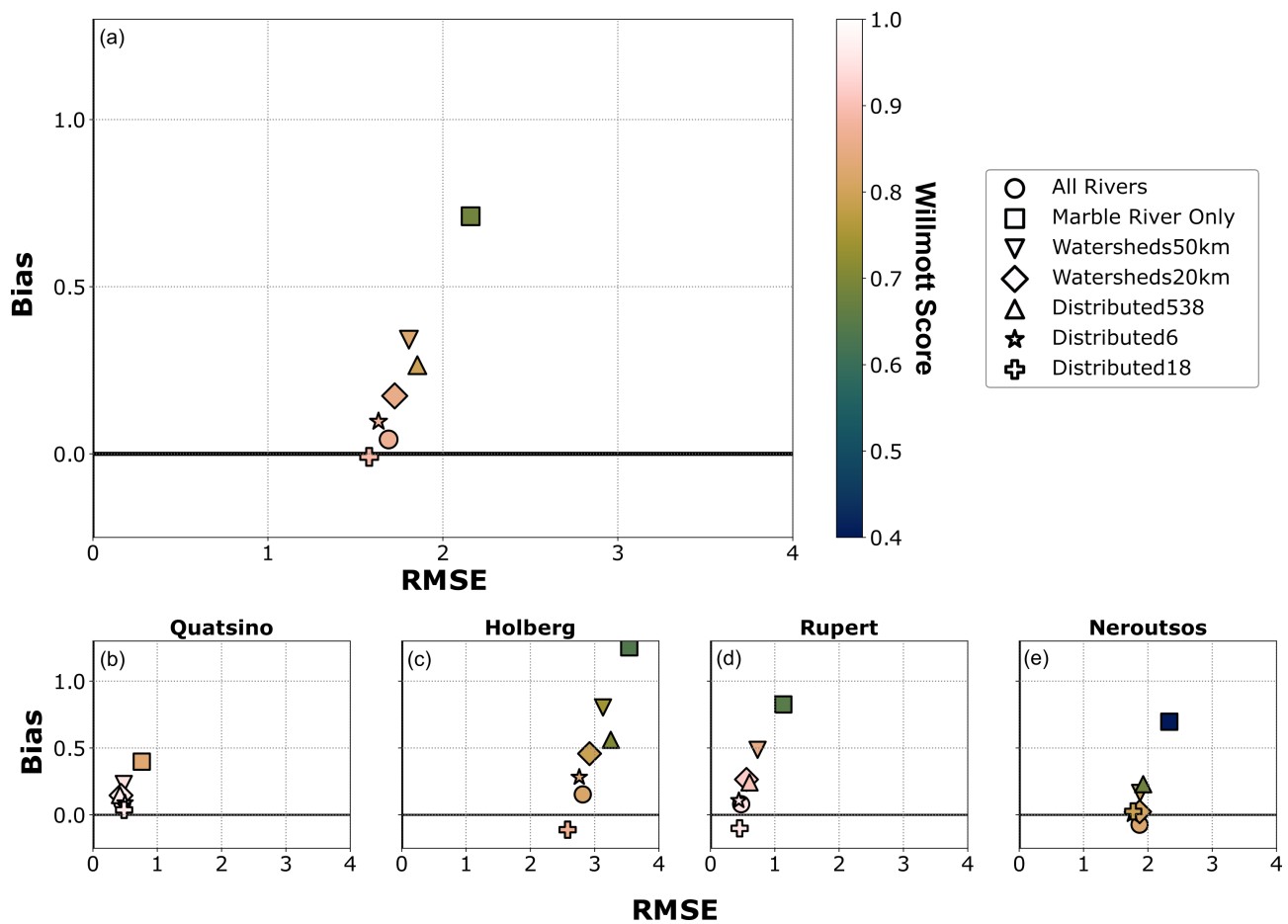

**Figure 9.** Summary of salinity model metrics for all sensitivity tests in the top 50 m. Bias is shown in the y axis, RMSE in the x axis, and Willmott Score in colour-scale. Model metrics are shown for (a) Entire model domain, (b) Quatsino Sound, (c) Holberg Inlet, (d) Rupert Inlet, (e) Neroutsos Inlet. Model metrics are also displayed in Table B1.

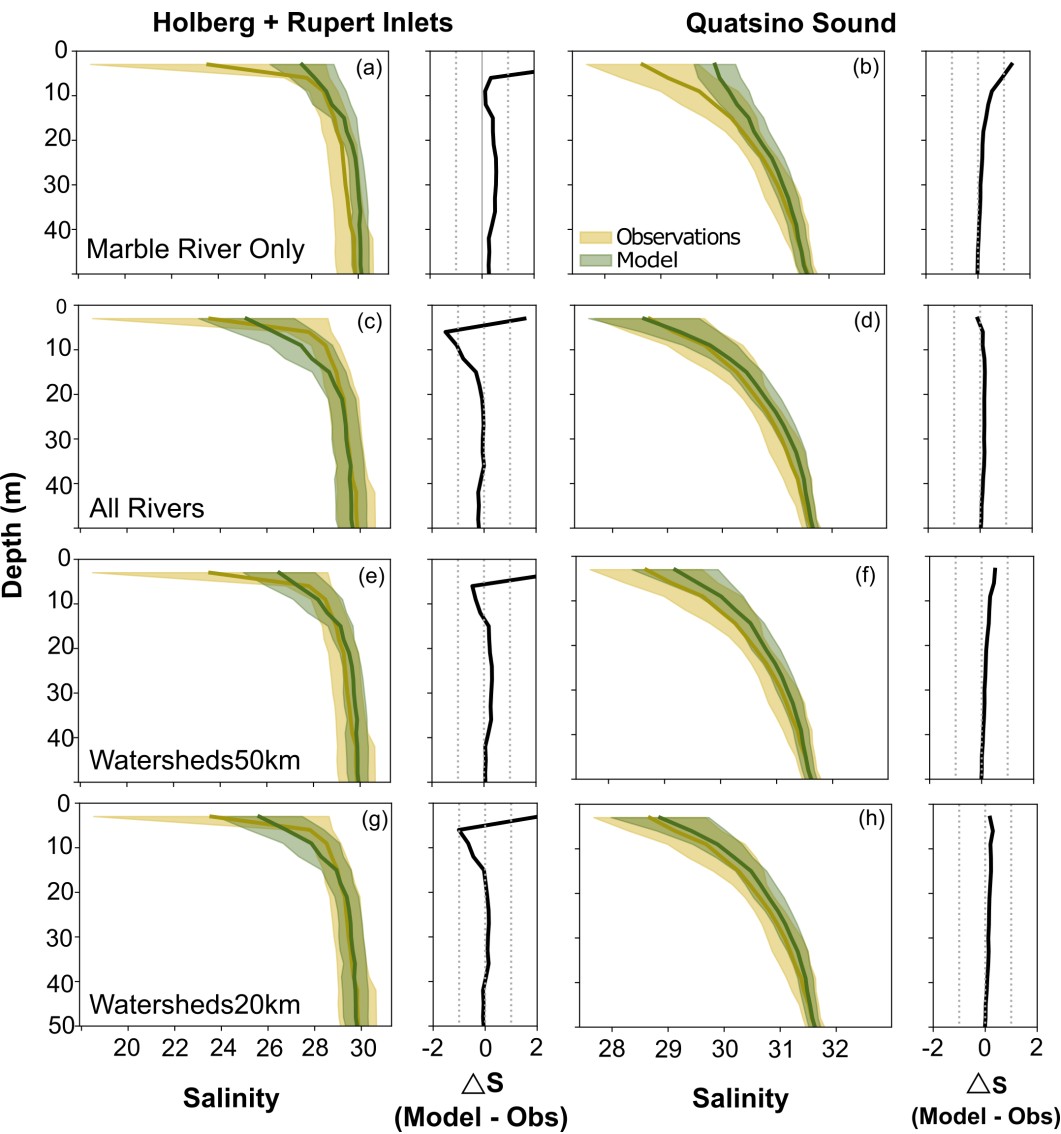

**Figure 10.** Mean profiles of salinity in the top 50 m in Holberg and Rupert Inlets combined (left) and Quatsino Sound (right) for (a,b) Marble River Only, (c,d) All Rivers, (e,f) Watersheds50km, and (g,h) Watersheds20km. In each plot, the left panel shows the temporal and spatial mean salinity profiles for the model (green) and observations (yellow) with the shaded area indicating one standard deviation. The right panel indicates the difference between the mean profiles.

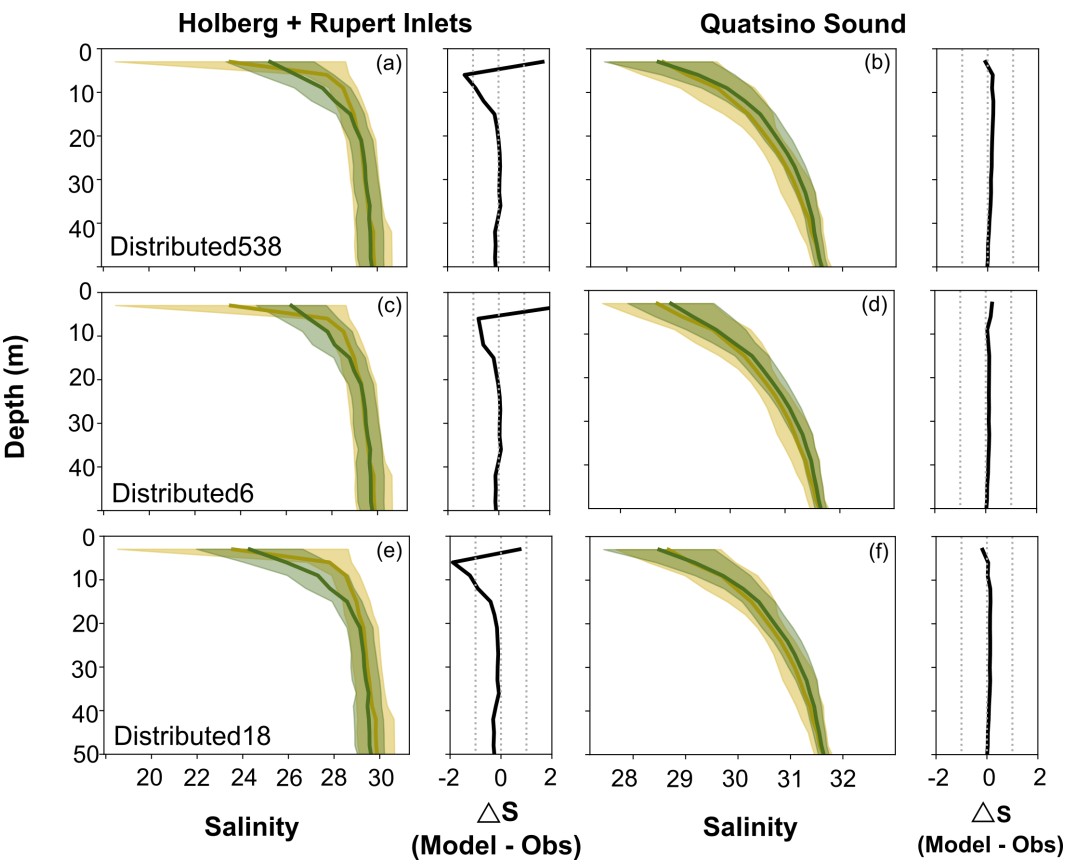

**Figure 11.** Mean profiles of salinity in the top 50 m in Holberg and Rupert Inlets combined (left) and Quatsino Sound (right) for (a,b) Distributed538, (c,d) Distributed6, and (e,f) Distributed18. In each plot, the left panel shows the temporal and spatial mean salinity profiles for the model (green) and observations (yellow) with the shaded area indicating the standard deviation. The right panel indicates the difference between the mean profiles.

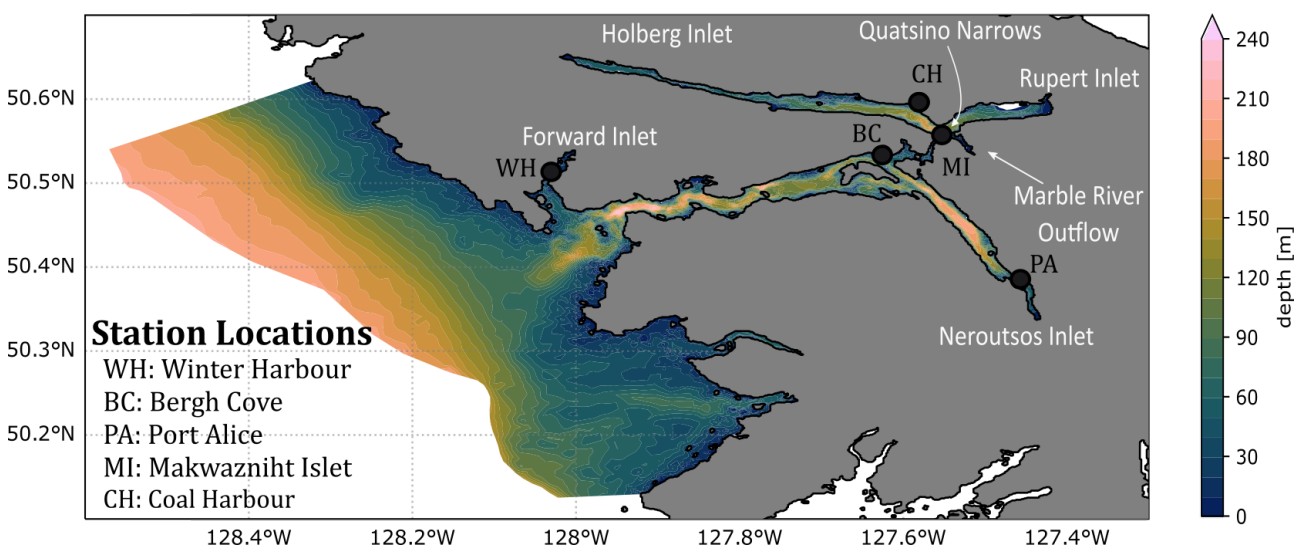

**Figure A1.** Map of modelled region, including 5 station locations measuring sea surface height: Winter Harbour (WH), Bergh Cove (BC), Port Alice (PA), Islet (MI), and Coal Harbour (CH).

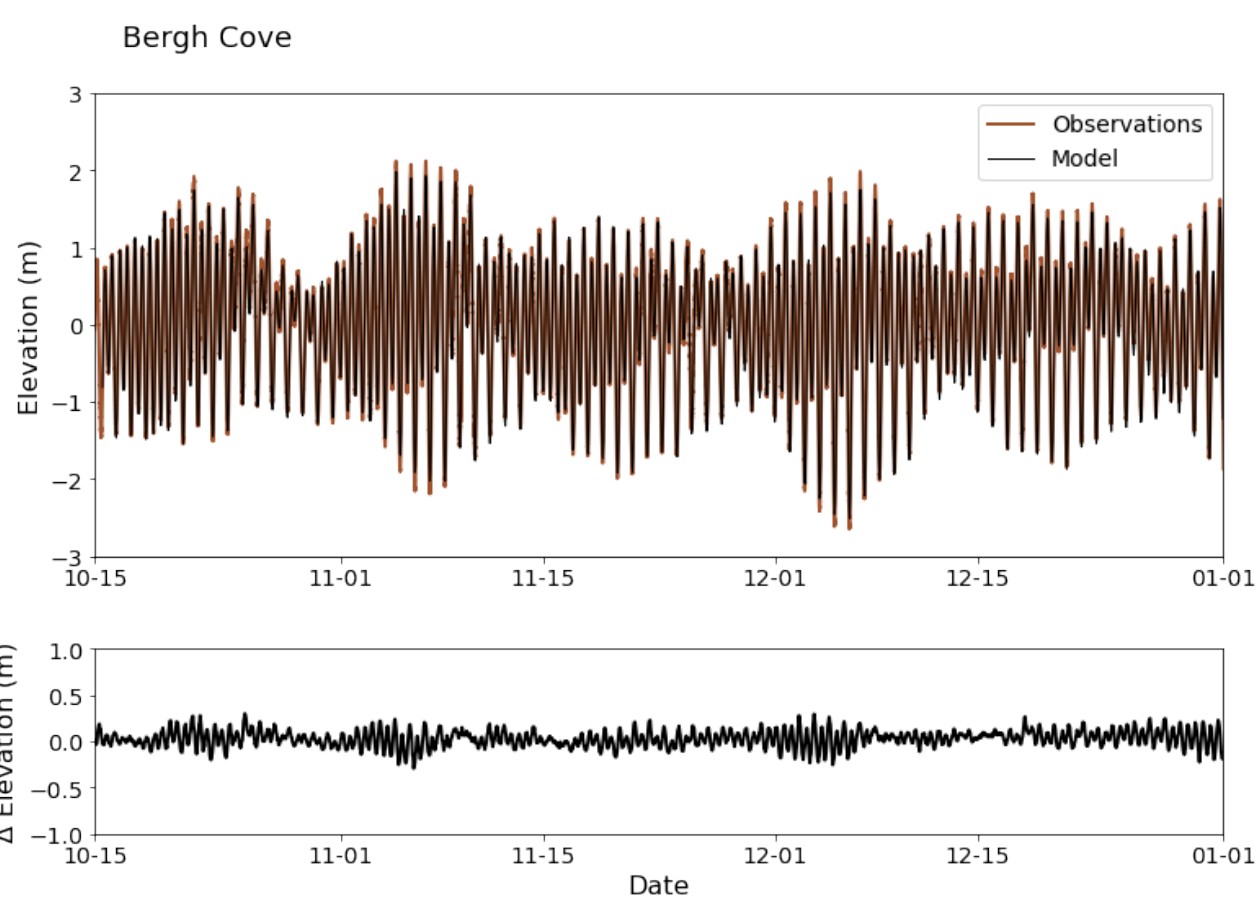

**Figure A2.** Top panel: Changes in elevation from the tidal gauge (red) located at Bergh Cove (see Figure A1) compared to the model (black) at the same location. Bottom panel: Difference between observations and model (observations minus model). Negative values indicate model overestimation.

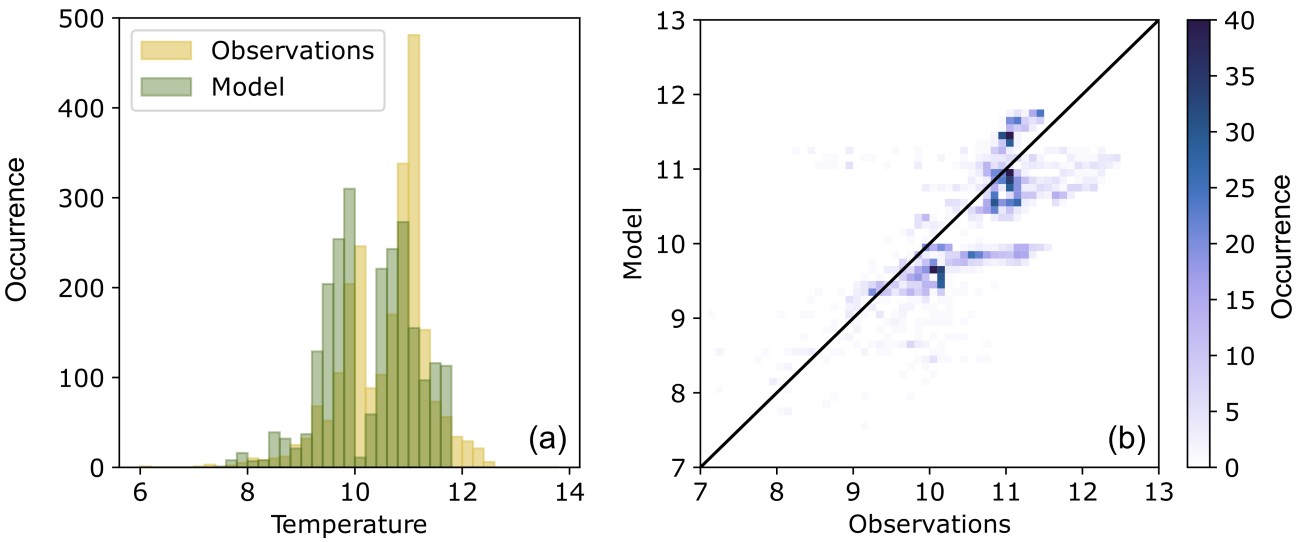

**Figure A3.** Temperature model evaluation for top 50 m in model: (a) 1D histograms, where yellow is the observed distribution of temperature and green is the modelled distribution of temperature ($^{o}$C); (b) 2D histograms of model (y axis) vs observations (x axis).

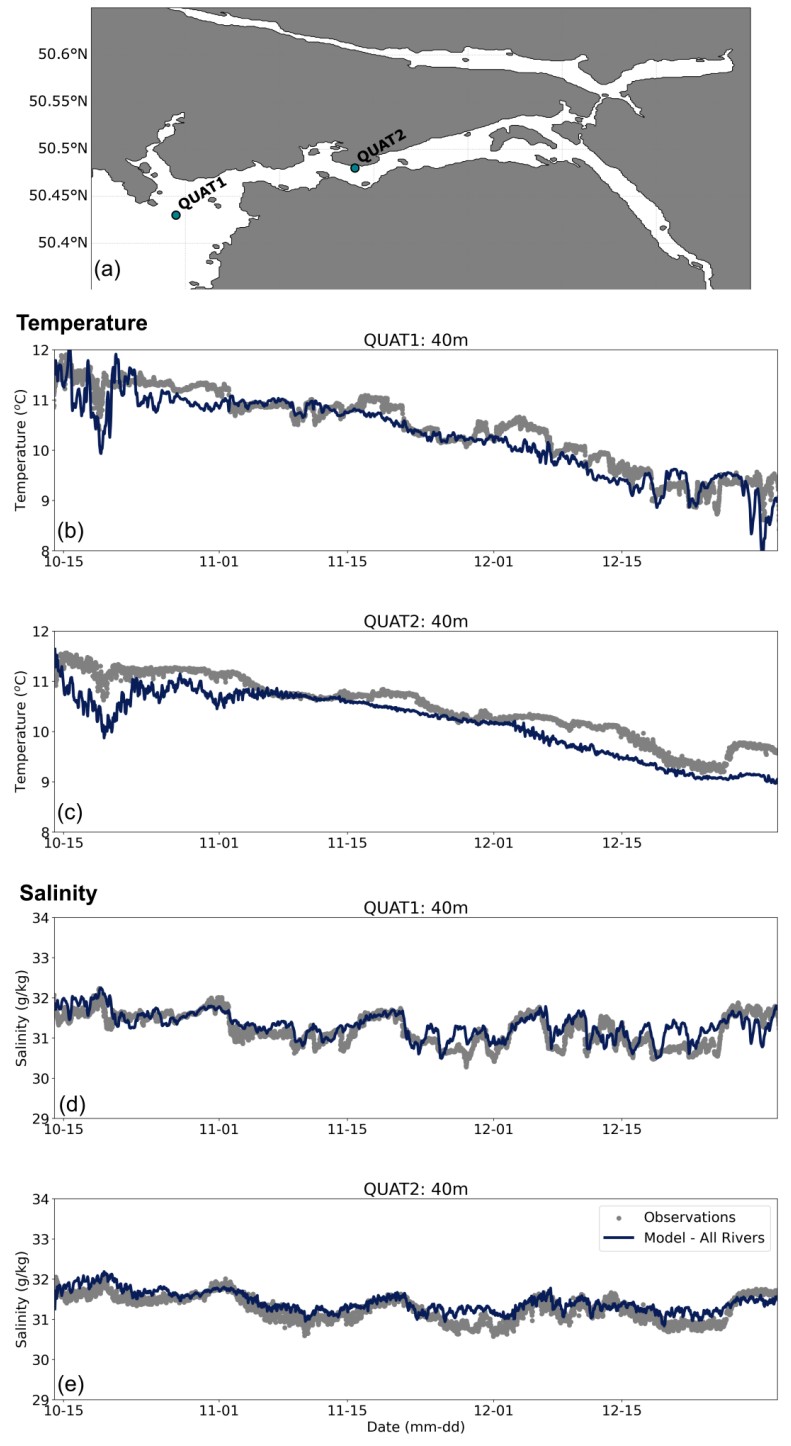

**Figure A4.** Comparison of modelled temperature ($^{o}$C, dark blue) to observed temperature ($^{o}$C, grey) at 40 m depth at (b) QUAT1 and (c) QUAT2; and modelled salinity (g/kg, dark blue) to observed salinity (g/kg, grey) at 40 m depth at (d) QUAT1 and (e) QUAT2. Mooring locations are indicated in the top panel (a).

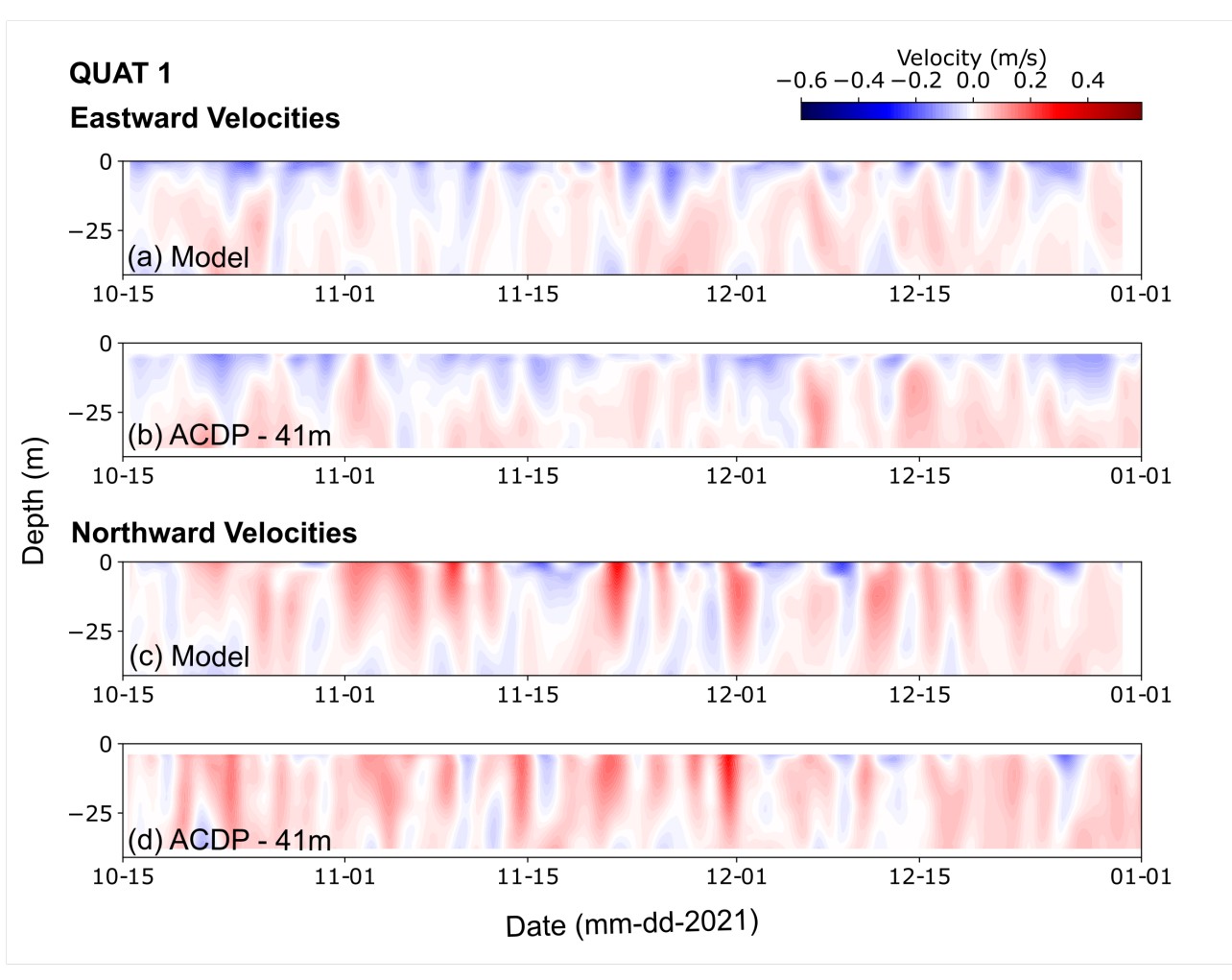

**Figure A5.** Eastward (a,b) and northward (c,d) velocities at QUAT1 mooring (location in Figures A4a) from the model (a,c) and ADCP at 41 m depth (b,d).

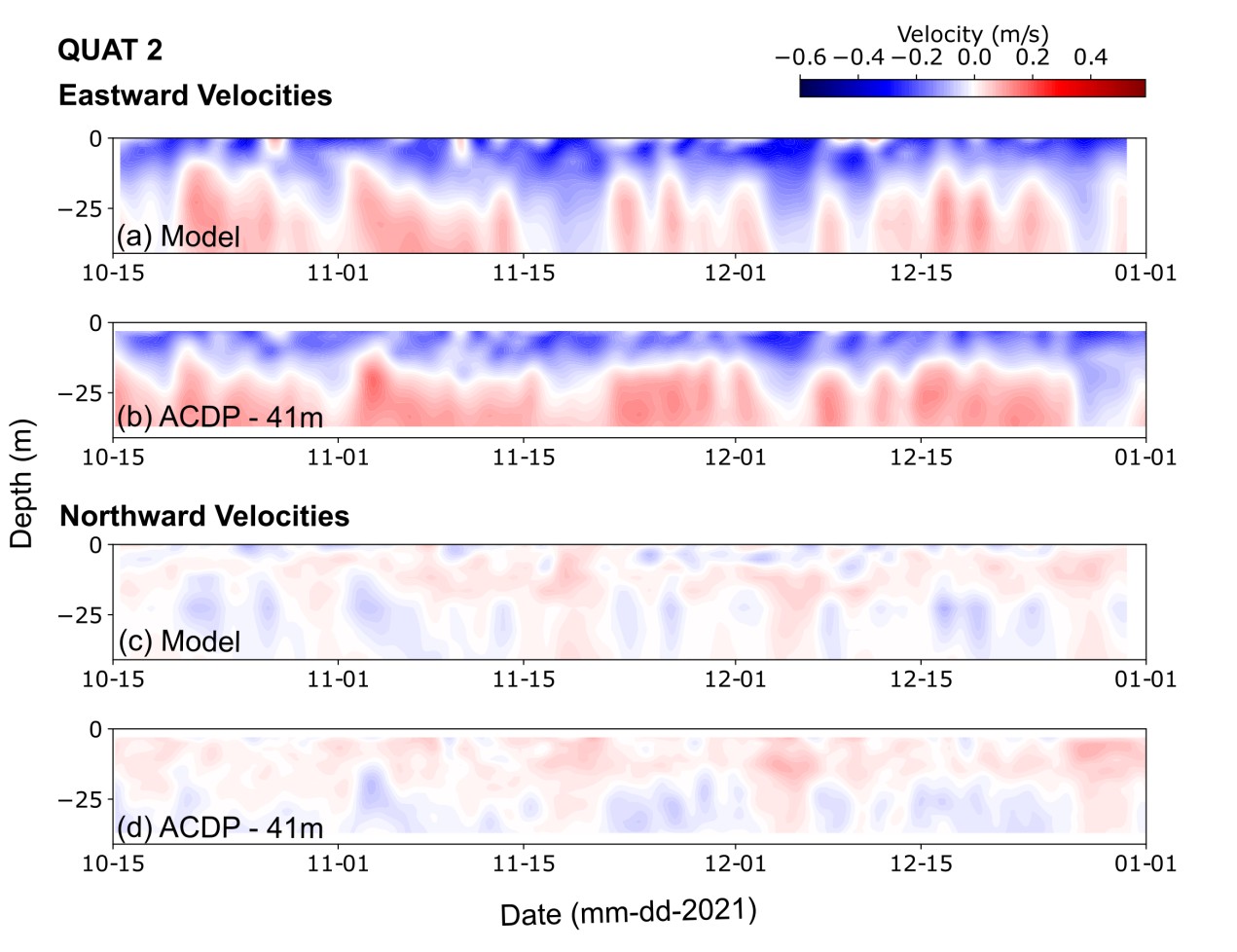

**Figure A6.** Eastward (a,b) and northward (c,d) velocities at QUAT2 mooring (location in Figures A4a) from the model (a,c) and ADCP at 41 m depth (b,d).

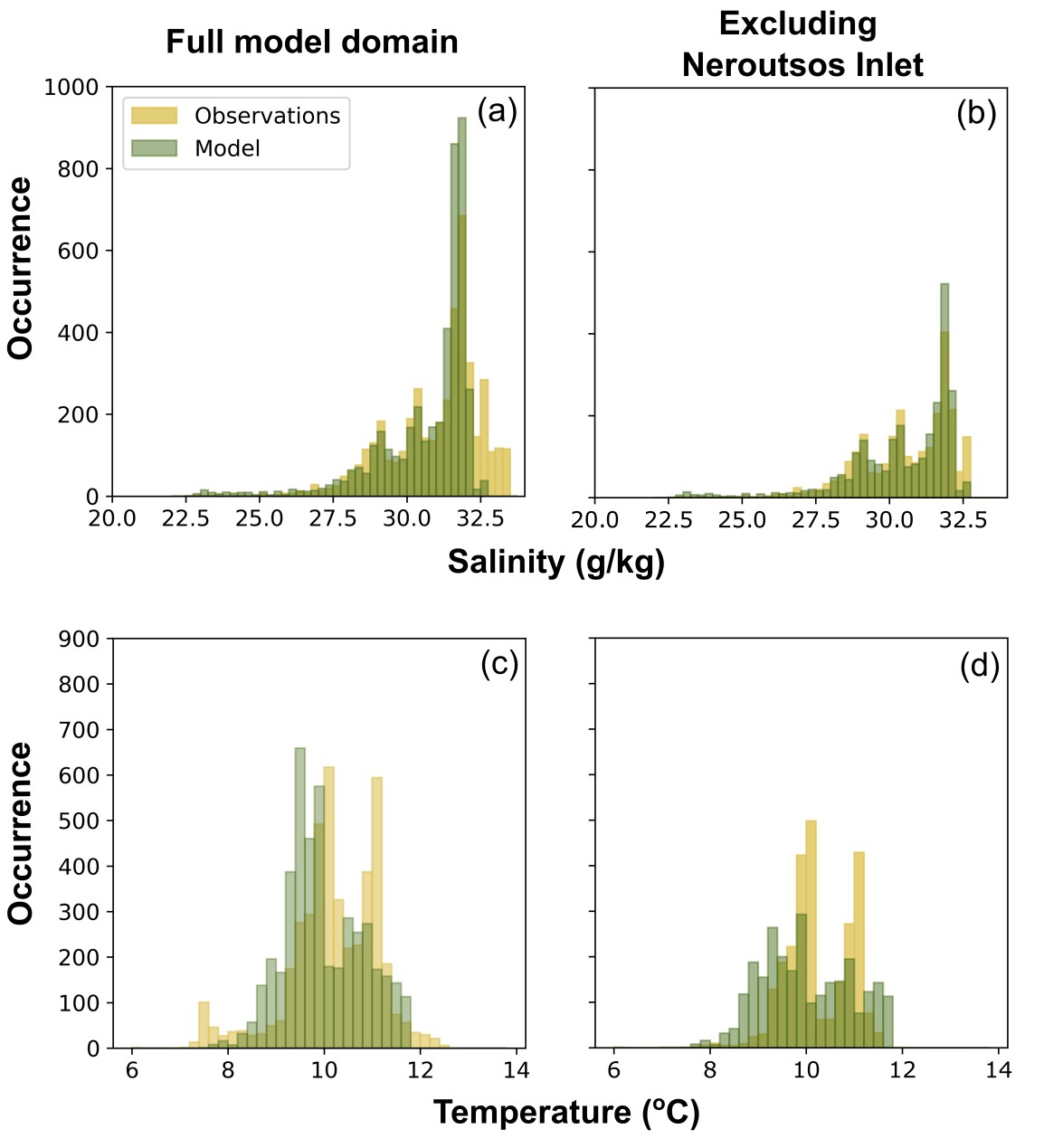

**Figure A7.** Histograms of (a,b) salinity (g/kg) and (c,d) temperature ($^o$C) for the model (green) and observations (yellow) for the entire water column. The left panels show histograms for the entire model domain whereas the right panels show the histograms for the model domain excluding Neroutsos Inlet.

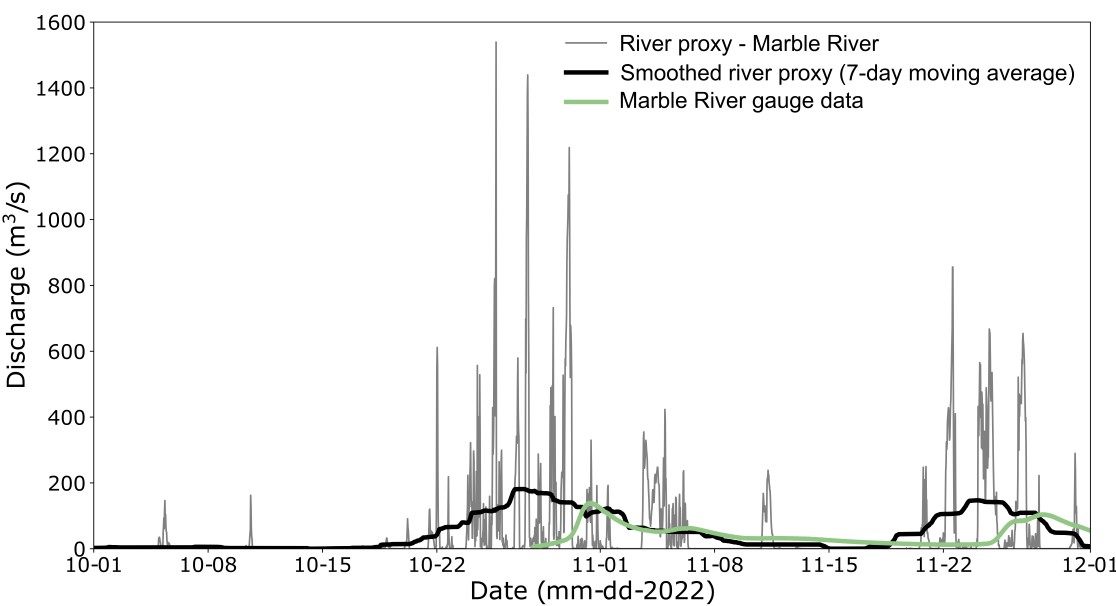

**Figure C1.** Comparisons of the rain-based proxy (grey) to the Marble River gauge (light green) discharge (m$^3$ s$^{-1}$) data for October and November 2022. A 7-day running mean of the proxy is shown in black. October and November 2022 are shown since this is the only period of overlap between the two time series.