# Peer review of "A simple approach to represent precipitation-derived freshwater fluxes into nearshore ocean models: an FVCOM4.1 case study of Quatsino Sound, British Columbia"

_EGUsphere, 2023_

## Author Comment (AC3)

Review of "A simple approach to represent precipitation-derived freshwater fluxes into nearshore ocean models: an FVCOM4.1 case study" by Rutherford et al.

Reviewer comments in black; Responses in blue.

General comments:

This manuscript studies an important technical issue of how to represent freshwater fluxes accurately and efficiently in high-resolution nearshore ocean models. For coarse-resolution models, freshwater inputs are normally incorporated by aggregating individual contributions from various watersheds into one source and specifying it at one or two model grid points. This approach is definitely undesirable for high-resolution models in which explicit depiction of disparate spatial scales is at high priority. Using Quatsino Sound as a case study site and employing a relatively simple rain-based hydrological model, the authors perform a series of sensitivity tests to answer the two research questions they proposed. The topic is interesting and scientifically important within the geoscientific model development. The results derived here can be extended to other coastal areas and, thus, are worthy of publication. However, I would suggest the authors to clarify certain points and make some statements in the text more accurate.

We thank the Reviewer for their thoughtful comments and feedback.

Specific comments:

1) I would suggest that the title of the paper be changed to "A simple approach to represent precipitation-derived freshwater fluxes into nearshore ocean models: a case study of Quatsino Sound, British Columbia".

This method is not unique to FVCOM. If you change to another coastal ocean circulation model, the proposed rain-based hydrological model can still be used. In the meantime, the conclusions of the paper, such as those listed in the Abstract, is only valid for Quatsino Sound.

We agree with the Reviewer that this is not unique to FVCOM; however, GMD requires that the model name be included in the title for this type of manuscript. We thus propose changing the title to:

*A simple approach to represent precipitation-derived freshwater fluxes into nearshore ocean models: an FVCOM4.1 case study of Quatsino Sound, British Columbia*

2) This paper only considers the very simple rain-based hydrological model. Actually, within the framework of FVCOM, a more accurate way to estimate the effect of precipitation-derived freshwater fluxes on fjord salinity dynamics can be done. In this approach, nearshore ocean model domain will be enlarged to encompass all the watersheds in the study area. Then using FVCOM's wetting and drying capability to simulate over land flow due to rain events using FVCOM's precipitation and evaporation forcing. Have the authors tried this approach? Of course, it requires a lot more computational time.

This is an interesting suggestion! While we have done some tests with wetting/drying, we have not specifically tried this approach for the implementation of freshwater fluxes into our model. The main goal of this paper is to present an approach that could be applied to other types of models and not just FVCOM; we thus feel doing further tests might be out of scope for this paper and propose potentially testing this approach in future iterations of the model. Additionally, in terms of our long-term goals, we plan to run the model for a longer period of time, and thus the computational expense of the proposed approach would likely make it disadvantageous.

Do the authors consider evaporation or evapotranspiration, in addition to precipitation?

Yes, the precipitation fields from HRDPS-1km factors in evaporation (i.e., it is precipitation minus evaporation) so it is inherently considered in our methods. We will clarify this in the text at line 157ff as follows (changes in ***bold italics***):

"Due to the lack of gauged rivers and streams, we developed a discharge proxy based on instantaneous precipitation ***minus evaporation*** from the HRDPS-1km atmospheric model for each of the 539 watersheds."

3) Lines 170-173, do you consider the freshwater falling on the surface of the numerical domain (i.e., the fjord system)? If not, the simulated salinity field will be biased.

Yes, precipitation/freshwater falling on the surface of the numerical ocean model domain is included. At line 136, we indicate which atmospheric conditions are included in the model configuration, and we will further clarify this at lines 170-173 as follows (changes in **_bold italics_**):

"The proxy-calculated river discharge is illustrated in Figure 4, highlighting how much **_river and stream_** water might be missing from the model inputs if only Marble River discharge was considered."

4) Lines 228-249, the authors should give a definition of "mean" or "average" in the paragraphs here. Otherwise, it will take the readers a lot of time to try to figure that out.

We understand the confusion. First of all, we will first make sure we consistently use the same term throughout this section of the text. We will additionally include the following definition at lines 216ff: "We calculated the arithmetic mean for most model-observation comparisons, which we will henceforth simply refer to as mean values."

5) Lines 297-298, "All other sensitivity tests had metrics in between those of the Marble River Only and All Rivers simulations". This is definitely a wrong statement, which is not consistent with the numbers (e.g., Willmott Score) quoted in the text. Fig. 9 is another source to check with.

Agreed. We will change this sentence to: "All other sensitivity tests had metrics with salinity bias less than 0.4 g/kg, RMSE less than 2g/kg and Willmott Score higher than 0.8."

6) Lines 310-313, to make sure this statement is correct, you can either use the general vertical coordinate in FVCOM simulation, and/or greatly increase the number of the vertical layers. Have you tried these?

We have tried different types of vertical coordinates as well as different numbers of vertical layers in both this domain and other model domains in the region, finding

consistently overly diffusive results in all cases. We are running further tests to hopefully minimize this effect in future iterations of the model. Additionally, we realize we were not specific in our methods section – at lines 110 we specify that we use terrain-following sigma coordinates in the vertical; we will update this line to clarify that we actually do use s-coordinates (or general vertical coordinates/tanh sigma coordinate type) with DU=3.0, DL=0.0 and a sigma power of 1.0.

Technical corrections:

1) Line 128, add "on" before October 14th, 2021

Thank you for catching this.

2) Line 131, add "on" before October 14th, or (it would be better) add "at 00:00 am on October 14th"

We will correct this.

3) Line 162, delete "in each watershed"

We will correct this.

4) Line 169, Equation (3). Should it be $A_{HRDPS,j}$ the denominator and $A_{ws,j}$ the numerator?

Thank you for catching this! $A_{HRDPS}$ should be the denominator and $A_{ws}$ should be the numerator – it is coded correctly but we accidentally reversed them in the equation in the manuscript. This will be corrected.

5) Line 319, add "as" after "as long"

We will correct this.

6) Figure 4 caption, "in equation 2" or in equation 3?

You're right, this should be equation 3. We will correct this.

7) Figure 6, for right-hand side panels I would suggest to change the color scale to blue color only because, I guess, no positive difference in surface salinity exists in the result.

We will modify this.

8) Figure 7 caption, is this called a histogram? I can understand that Figure 8 is called a histogram, but not this one.

This is a 2D histogram or it may also be referred to as a density heat map. 2D histograms can be used instead of scatter plots when there is a lot of overlapping data, such as in our model-observation comparisons, to indicate where there is a higher density of points. I specifically use the function (in Python) matplotlib.pyplot.hist2d to create the figures in Figure 7.

9) Figure 9, a Table may be a better choice than a Figure

We argue that a figure is quite informative, particularly for visual people. However, we understand that people process information differently and we can certainly add a table with the same information in the appendix.

---

## Author Comment (AC4)

Review of "A simple approach to represent precipitation-derived freshwater fluxes into nearshore ocean models: an FVCOM4.1 case study" by Rutherford et al.

Reviewer comments in black; Responses in blue.

General comments:

The paper, "A Simple Approach to represent precipitation-derived Freshwater Fluxes into Nearshore Ocean Models: an FVCOM4.1 Case Study," presents a straightforward solution that significantly improves the representation of hydrological influence in ocean circulation models. The authors demonstrate a keen understanding of the simplicity of this solution that would be beneficial to oceanographers when improving the results of their model. For example, the decision to utilize the grid of the atmospheric model HRDPS aligned well with this principle. While not perfect, this approach eliminates the need for remapping operations and mitigates other potential sources of error, such as water conservation.

Furthermore, the authors explore various gaps in the knowledge of the river network, including the watershed area and river mouth location. They provide valuable insights into where efforts should be concentrated to enhance the efficiency of coastal circulation models. Despite these strengths, the authors remain cognizant of the existing gaps in their rivers model.

We thank the Reviewer for their thoughtful comments and feedback.

Specific comments:

- The river proxy currently relies solely on precipitation data. Incorporating evaporation data (P-E) could enhance the proxy's accuracy in predicting the volume of water in the river system.

We agree that adding evaporation into the calculation would enhance the proxy's accuracy and will add a discussion of this in the manuscript. Nevertheless, evaporation was negligible compared to precipitation in the region during our simulation period such that it would have had a minimal role in our simulations.

The following text will be added to the manuscript (new text in **bold italics**):

Line 170: The proxy-calculated river discharge is illustrated in Figure 4, highlighting how much fresh water might be missing from the model inputs if only Marble River discharge was considered. ***While the proxy currently only considers precipitation, the method could benefit from using precipitation minus evaporation instead to enhance the proxy's accuracy; however, evaporation was minimal during the modelled period compared to precipitation in the region and was not included in the current work***. ***Evaporation would become more important during the spring and summer***. For inputting into the model, river salinity was set to zero and river temperature was set equal to the temperature time series from the nearby Nimpkish River (Water Survey of Canada, 2023a) for all rivers and streams.

Line 357: The only requirements to estimate river and stream runoff through this approach are (1) a rudimentary knowledge of watershed area and, ideally, outpour locations, and (2) precipitation ***(or precipitation minus evaporation)*** from an atmospheric model.

- The frequency of the river proxy used in the FVCOM model is unclear. While some analyses were conducted at a high frequency (as shown in Figure 4), others were done on a monthly basis (Figure 5). Clarification on the frequency of these inputs implemented in the model would be beneficial.

We will add clarification on this at lines 171 (additions in ***bold italics***):

"For inputting into the model, river salinity was set to zero and river temperature was set equal to the temperature time series from the nearby Nimpkish River (Water Survey of Canada, 2023a) for all rivers and streams***. All river forcing variables (temperature, salinity and proxy-calculated river flux) had a forcing frequency of 30 minutes (as shown in Figure 4).*** "

- Insufficient time was dedicated to validating the CIOPS-W model in the region under study. This raises concerns about whether the CIOPS-W model itself could be a source of errors from the initial states and the boundary conditions used in their coastal circulation model.

We found that the Quatsino Sound model performs well at the outer mooring (QUAT1; Figure A4a in manuscript), which experiences the effects from both CIOPS-W's initial and open boundary conditions. To illustrate this point, we show comparisons of the model to temperature and salinity data from QUAT1 at two depths (40 and 100 m; Figure R2.1 below). The model also performs well at QUAT2, which is farther in the inlet

and consequently only sees effects of CIOPS-W open boundary conditions (initial conditions here are based on CTD observations). We thus feel confident that CIOPS-W provides appropriate initial and boundary conditions on the shelf for the present study. That said, the current manuscript focuses on the upper 50 m of the water column inside the Sound, where the role of the open boundaries is minimal due to the typical estuarine circulation. Therefore, we do not plan to add extra material in this study to provide confidence into CIOPS-W-based initial and boundary conditions; however, we will make use of this useful comment and address this topic in our next manuscript.

[Figure]

*Figure R2.1: Comparisons of temperature and salinity from the model vs observations from QUAT1 mooring at 40 and 100 meters. This figure illustrates the model's performance over the shelf, near the mouth of the Sound, highlighting how well CIOPS-W works as initial and open boundary conditions. The time series at 40 m are also shown in Figure A4 in the manuscript.*

- The paper frequently mentions that the river proxy is rain-based and does not account for snow coverage. Conducting a climatological overview of the region, particularly regarding winter snow coverage or snowfall, could help assess the extent of this limitation in the current model.

We will add more details about the seasonal snow coverage. Bidlack et al. (2021) describes the watershed type for many watersheds along the Pacific Northwest of

North America, and their results are based on 30 years of data from the Distributed Climate Water Balance Model. The data shown in their Figure 4 indicates that, averaged over 1981-2010, the Marble River and 4 smaller watersheds that drain into Quatsino Sound can be categorized as hybrid snow- and rain-driven runoff regime watersheds. This means that, on average, these 5 watersheds experience a snow melt signal in spring and elevated flow in fall/early winter from rain and rain-on-snow events. The remaining 534 watersheds in our study are rain-dominated regimes, with occasional transient snowpacks contributing minimally to stream flows . We will add a few sentences briefly explaining these results from Bidlack et al. at around lines 179-182. This will be in addition to the text we already have at lines 180-182 describing the snow coverage during the simulation period.

Bidlack, A. L., Bisbing, S. M., Buma, B. J., Diefenderfer, H. L., Fellman, J. B., Floyd, W. C., Giesbrecht, I., Lally, A., Lertzman, K. P., Perakis, S. S., et al.: Climate-mediated changes to linked terrestrial and marine ecosystems across the Northeast Pacific coastal temperate rainforest margin, BioScience, 71, 581–595, https://doi.org/10.1093/biosci/biaa171, 2021.

- Validation of the properties of deeper water should be considered. While the 0-50m depth is directly affected by river flow, this could also influence the content of deeper water through estuarine circulation.

While we agree in principle with this comment, we think that adding detailed validation of the full water column to the present manuscript will make it too long and is slightly out of scope given the focus on the top 50 m (i.e., the region where freshwater will have the largest impact).  We intend to include a more detailed, whole-column validation in an upcoming manuscript that will focus on all depths in the modelled region. We thus propose adding the figure and text below to the appendix as a brief evaluation of the model's performance for the whole water column.

**Appendix A5 CTD profiles: Full Water Column Evaluation**

Additional evaluation of modelled salinity and temperature against CTD profiles spanning the full water column showed reasonable agreement (Figure A6). The distribution of modelled salinity agrees well with the CTD observations (Figure A6a), particularly if we exclude Neroutsos Inlet from the analysis (Figure A6b). As discussed, the model shows over-mixed conditions in this deep and narrow inlet (see more details in lines 307-313). Modelled temperature adequately compares to CTD temperature observations throughout the entire water column (Figure A6c), with a similar comparison as in the top 50 m (Figure A3a). More specifically, the model has a smaller temperature range than the observations, unable to capture the cooler (<

*8ºC) and warmer (>12ºc) temperatures, in large part due to the issues encountered in Neroutsos Inlets (Figure A6d). Future work will perform a more rigorous evaluation of the full water column.*

[Figure]

*Figure A6: Histograms of (a,b) salinity (g/kg) and (c,d) temperature (ºC) for the model (green) and observations (yellow) for the entire water column. The left panels show histograms for the entire model domain whereas the right panels show histograms for the model domain excluding Neroutsos Inlet.*

- The modelling of water storage will primarily delay and temper peak precipitation, rather than limit the total volume of water flowing into the river system (line 176).

*We will modify this sentence accordingly to clarify this point (modifications in **bold italics**).*

*"There will be some storage of the precipitation (e.g. in soil and lakes), which **would primarily delay and** limit the **peak** amount of rain that **discharges into the fjord system**; in the particular case of the Marble River, the lakes within the watershed will partly store some of the precipitation entering this watershed and slow the water as it flows through the system."*

- A comparison with the Marble River at a higher time resolution than monthly could provide more detailed insights (line 196). One suggestion could be to combine Figures 4 and 5 for this purpose.

Rather than merging Figures 4 and 5, we will add a new figure in the appendix to provide further visuals to compare the proxy to the Marble River gauge data at higher resolutions. As we mention at line 345, the assumptions in this method produce at times large spikes or pulses in river discharge, most notably for the Marble River (seen in Figure 4). These spikes are unrealistic when compared to hourly discharge time series from most large rivers, which are typically much smoother. That said, the proxy is able to represent the magnitude of the discharge on longer timescales, for example monthly, as seen in Figure 5, and weekly, as it will be shown in the new Appendix B figure below.

Additionally, while we have discussed these limitations in the Discussion (lines 344-345), we will further emphasize them in the methodology section at lines 190ff as follows (main changes in **bold italics**; editorial/style edits not highlighted here):

"Despite these limitations, the simplicity and ease of application of the method make it worthy of analysis, as it could provide some information on river and stream outflows that may otherwise be unavailable to and thus ignored by coastal ocean modellers. To illustrate the proxy's ability to estimate river discharge, we evaluated the performance of these methods for the Marble River; the river proxy was applied to the available HRDPS-1km precipitation time series (2021-2022) for comparison against the available discharge data, which has only been collected since October 2022. Although these time periods have minimal overlap ($\sim$ 2 months), the available data can inform the proxy's ability to estimate river discharge. For instance, the monthly mean river discharge estimated by the river proxy is comparable to the monthly mean discharge measured by the Marble River gauge (Figure 5). Notably, the 2021 river proxy values for September to November were higher than values for the same period in 2022 calculated from both the proxy and from the river gauge. Considering that fall 2021 was a particularly rainy year that included an atmospheric river in mid-November, these values are reasonable. **Furthermore, while the half-hourly proxy dataset shows large spikes, a 7-day running mean suggests that the method produces an appropriate amount of river discharge at**

*weekly timescales (Figure B1)*. Overall, these comparisons illustrated the proxy's ability to reasonably represent the freshwater discharge at weekly and longer scales.

**Appendix B: Additional evaluation of the rain-based river proxy**

*To further illustrate the ability of the rain-based proxy to provide reasonable amounts of fresh water to the Quatsino Sound fjord system, we show comparisons of the proxy (applied to the Marble River watershed), its 7-day running mean, and the Marble River gauge data for October and November 2022 (Figure B1). The proxy and gauge data only overlap for this short period at the end of 2022. While the proxy time series (grey; datapoints every 30 minutes) shows large spikes due to its assumptions and limitations (see Section 3.3.3), the 7-day running mean of the proxy (black) produces more realistic values in alignment with the data from the Marble River gauge (light green). While the magnitude of the smoothed proxy is similar to the Marble River gauge data, the limitations of the methodology (e.g., no storage of rain in soil or lakes) are still evident. For instance, the 7-day running mean of the proxy produces river discharge that peaks earlier than the gauge data during rain events (e.g. November 20-30th). Additionally, the smoothed proxy time series reaches a minimum value of 0 $m^3$/s in between rain events whereas the gauge data does not (see around November 15th). Despite these differences, the proxy is able to deliver adequate amounts of fresh water to the system over a 7-day period.*

[Figure]

*Figure B1: Comparisons of the rain-based proxy (grey) to the Marble River gauge (light green) discharge ($m^3/s$) data for October and November 2022. A 7-day running mean of the proxy is shown in black. October and November 2022 are shown since this is the only period of overlap between the two timeseries.*

- The paper states that the All Rivers configuration performs better, but then suggests that only 60-75% of the total freshwater source is necessary (line 324). An explanation of this apparent contradiction would be helpful.

We will add clarification in the paragraph including line 324 (see below). What we want to convey is that we realize that it is not always realistic/possible to include all watersheds in coastal models, but according to our tests, including at least 60-75% of total freshwater fluxes will provide similar results to a model simulation that includes all watersheds. Although these specific numbers are based on our own region, we believe that the general principle could be applied to other similar regions. Another take-home message that we want to emphasize is that the common approach of only including the major rivers (exemplified by our Marble River Only simulation) is not enough, such that efforts need to be made to add at least 60% of the incoming freshwater.

Lines 317ff (additions in ***bold italics***; modifications in **bold**): Although watershed analyses established that 539 rivers and streams flow into the Quatsino Sound fjord system, ***our*** results indicate that ***good model performance can be achieved*** as long ***as*** 60% or more of the fresh water is accounted for. ***Compared with the Marble River Only simulation,*** **m**odel performance improved just by including rivers with watersheds greater than 50 km$^2$ (7 rivers total and ~58% of total flux; Figures 7-10) and including rivers with watersheds greater than 20 km$^2$ improved the model results even further

(19 rivers total and ~ 75% of total flux; Figures 7-10). The latter sensitivity test performed similarly to the All Rivers simulation (Figures 6 - 10); however, it is worth noting that the Watersheds20km simulation had a larger salinity bias than the All Rivers case in the upper 50 m in both Holberg (0.5 vs. 0.2 g/kg) and Rupert (0.25 vs. <0.1 g/kg) Inlets (Figure 9). We conclude ***that including only the major rivers is not enough to achieve good coastal model performance (e.g., our Marble River Only simulation). Our results also suggest that if watershed information is limited, including all freshwater sources is likely not necessary for improved model performance;*** in our case study, including at least 60-75% of total freshwater sources substantially improved ***upon the Marble River Only simulation***. While the exact amount of freshwater discharge required will depend on the specific system being modelled, these results might be useful for guiding model development for other regions.

---

## Author Comment (AC5)

Addendum/Corrigendum to Reviewer 1's question "*Do the authors consider evaporation or evapotranspiration, in addition to precipitation?*"

At the time of the original response, we thought that the atmospheric model was providing net precipitation (P-E). However, we have now confirmed that evaporation is not included within our precipitation fields, such that it is not considered in our proxy. That said, E is negligible compared to P in our region and period of simulations, such that E would have a minimal role.

Instead of the modification to the text suggested in our first response to Reviewer 1, we now plan to add the following text:

Line 170: The proxy-calculated river discharge is illustrated in Figure 4, highlighting how much fresh water might be missing from the model inputs if only Marble River discharge was considered. ***While the proxy currently only considers precipitation, the method could benefit from using precipitation minus evaporation instead to enhance the proxy's accuracy; however, evaporation was minimal during the modelled period compared to precipitation in the region and was not included in the current work***. ***Evaporation would become more important during the spring and summer.*** For inputting into the model, river salinity was set to zero and river temperature was set equal to the temperature time series from the nearby Nimpkish River (Water Survey of Canada, 2023a) for all rivers and streams.

Line 357: The only requirements to estimate river and stream runoff through this approach are (1) a rudimentary knowledge of watershed area and, ideally, outpour locations, and (2) precipitation ***(or precipitation minus evaporation)*** from an atmospheric model.

---

## Author Response (AR1)

**Response to Review 1** of "A simple approach to represent precipitation-derived freshwater fluxes into nearshore ocean models: an FVCOM4.1 case study" by Rutherford et al.

Reviewer 1 comments in black; Responses in blue.

General comments:

This manuscript studies an important technical issue of how to represent freshwater fluxes accurately and efficiently in high-resolution nearshore ocean models. For coarse-resolution models, freshwater inputs are normally incorporated by aggregating individual contributions from various watersheds into one source and specifying it at one or two model grid points. This approach is definitely undesirable for high-resolution models in which explicit depiction of disparate spatial scales is at high priority. Using Quatsino Sound as a case study site and employing a relatively simple rain-based hydrological model, the authors perform a series of sensitivity tests to answer the two research questions they proposed. The topic is interesting and scientifically important within the geoscientific model development. The results derived here can be extended to other coastal areas and, thus, are worthy of publication. However, I would suggest the authors to clarify certain points and make some statements in the text more accurate.

We thank the Reviewer for their thoughtful comments and feedback.

Specific comments:

1) I would suggest that the title of the paper be changed to "A simple approach to represent precipitation-derived freshwater fluxes into nearshore ocean models: a case study of Quatsino Sound, British Columbia".

This method is not unique to FVCOM. If you change to another coastal ocean circulation model, the proposed rain-based hydrological model can still be used. In the meantime, the conclusions of the paper, such as those listed in the Abstract, is only valid for Quatsino Sound.

We agree with the Reviewer that this is not unique to FVCOM; however, GMD requires that the model name be included in the title for this type of manuscript. We thus have changed the title to:

*A simple approach to represent precipitation-derived freshwater fluxes into nearshore ocean models: an FVCOM4.1 case study of Quatsino Sound, British Columbia*

2) This paper only considers the very simple rain-based hydrological model. Actually, within the framework of FVCOM, a more accurate way to estimate the effect of precipitation-derived freshwater fluxes on fjord salinity dynamics can be done. In this approach, nearshore ocean model domain will be enlarged to encompass all the watersheds in the study area. Then using FVCOM's wetting and drying capability to simulate over land flow due to rain events using FVCOM's precipitation and evaporation forcing. Have the authors tried this approach? Of course, it requires a lot more computational time.

This is an interesting suggestion! While we have done some tests with wetting/drying, we have not specifically tried this approach for the implementation of freshwater fluxes into our model. The main goal of this paper is to present an approach that could be applied to other types of models and not just FVCOM; we thus feel doing further tests might be out of scope for this paper and propose potentially testing this approach in future iterations of the model. Additionally, in terms of our long-term goals, we plan to run the model for a longer period of time, and thus the computational expense of the proposed approach would likely make it disadvantageous.

Do the authors consider evaporation or evapotranspiration, in addition to precipitation?

We agree that adding evaporation or evapotranspiration into the calculation would enhance the proxy's accuracy and have added a discussion of this in the manuscript. Nevertheless, we estimate that both evaporation and evapotranspiration were negligible compared to precipitation in the region during our simulation period such that either would have had a minimal role in our simulations.

The following text was added to the manuscript (new text in ***bold italics***):

Line 178ff in the revised manuscript: The proxy-calculated river discharge is illustrated in Figure 4, highlighting how much fresh water might be missing from the model inputs if only Marble River discharge was considered. ***While the proxy currently only considers precipitation, the method could benefit from using precipitation minus evapotranspiration to enhance its accuracy; however, evapotranspiration in the region was estimated to be minimal during the modelled period compared to***

*precipitation and was not included in the current work. Evapotranspiration would become more important during the spring and summer in the region.* For inputting into the model, river salinity was set to zero and river temperature was set equal to the temperature time series from the nearby Nimpkish River (Water Survey of Canada, 2023a) for all rivers and streams.

Line 378 in revised manuscript: The only requirements to estimate river and stream runoff through this approach are (1) a rudimentary knowledge of watershed area and, ideally, outpour locations, and (2) precipitation *(or precipitation minus evapotranspiration or evaporation)* from an atmospheric model.

3) Lines 170-173, do you consider the freshwater falling on the surface of the numerical domain (i.e., the fjord system)? If not, the simulated salinity field will be biased.

Yes, precipitation/freshwater falling on the surface of the numerical ocean model domain is included. At line 138, we indicate which atmospheric conditions are included in the model configuration, and we further clarified this at lines 177-178 as follows (changes in *bold italics*):

"The proxy-calculated river discharge is illustrated in Figure 4, highlighting how much *river and stream* water might be missing from the model inputs if only Marble River discharge was considered."

4) Lines 228-249, the authors should give a definition of "mean" or "average" in the paragraphs here. Otherwise, it will take the readers a lot of time to try to figure that out.

We understand the confusion. First of all, we will first make sure we consistently use the same term throughout this section of the text. We additionally included the following definition at lines 233ff: "We calculated the arithmetic mean for most model-observation comparisons, which we will henceforth simply refer to as mean values."

5) Lines 297-298, "All other sensitivity tests had metrics in between those of the Marble River Only and All Rivers simulations". This is definitely a wrong statement, which is not consistent with the numbers (e.g., Willmott Score) quoted in the text. Fig. 9 is another source to check with.

Agreed. We changed this sentence to: "All other sensitivity tests had metrics with salinity bias less than 0.4 g/kg, RMSE less than 2g/kg and Willmott Score higher than 0.8."

6) Lines 310-313, to make sure this statement is correct, you can either use the general vertical coordinate in FVCOM simulation, and/or greatly increase the number of the vertical layers. Have you tried these?

We have tried different types of vertical coordinates as well as different numbers of vertical layers in both this domain and other model domains in the region, finding consistently overly diffusive results in all cases. We are running further tests to hopefully minimize this effect in future iterations of the model. Additionally, we realize we were not specific in our methods section – at lines 110 we specify that we use terrain-following sigma coordinates in the vertical; we updated this line to clarify that we actually do use s-coordinates (or general vertical coordinates/tanh sigma coordinate type) with DU=3.0, DL=0.0.

Line 110ff (additions in **bold italics**): "The unstructured, triangular grid has 95,651 nodes and 181,696 elements horizontally, and uses **s-**coordinates in the vertical with 20 layers ***(also referred to as general vertical coordinates or tanh sigma coordinate type; upper and lower depth boundary parameters selected as DU=3.0 and DL=0.0, respectively***)."

Technical corrections:

1) Line 128, add "on" before October 14th, 2021

Thank you for catching this.

2) Line 131, add "on" before October 14th, or (it would be better) add "at 00:00 am on October 14th"

We corrected this.

3) Line 162, delete "in each watershed"

We corrected this.

4) Line 169, Equation (3). Should it be $A_{HRDPS,j}$ the denominator and $A_{WS,j}$ the numerator?

Thank you for catching this! $A_{HRDPS}$ should be the denominator and $A_{WS}$ should be the numerator – it is coded correctly but we accidentally reversed them in the equation in the manuscript. This is now corrected.

5) Line 319, add "as" after "as long"

We corrected this.

6) Figure 4 caption, "in equation 2" or in equation 3?

You're right, this should be equation 3. We corrected this.

7) Figure 6, for right-hand side panels I would suggest to change the color scale to blue color only because, I guess, no positive difference in surface salinity exists in the result.

We modified this.

8) Figure 7 caption, is this called a histogram? I can understand that Figure 8 is called a histogram, but not this one.

This is a 2D histogram or it may also be referred to as a density heat map. 2D histograms can be used instead of scatter plots when there is a lot of overlapping data, such as in our model-observation comparisons, to indicate where there is a higher density of points. We specifically use the function (in Python) matplotlib.pyplot.hist2d to create the figures in Figure 7.

9) Figure 9, a Table may be a better choice than a Figure

We argue that a figure is quite informative, particularly for visual people. However, we understand that people process information differently and we thus added a table with the same information to Appendix B.

**Response to Review 2** of "A simple approach to represent precipitation-derived freshwater fluxes into nearshore ocean models: an FVCOM4.1 case study" by Rutherford et al.

Reviewer 2 comments in black; Responses in blue.

General comments:

The paper, "A Simple Approach to represent precipitation-derived Freshwater Fluxes into Nearshore Ocean Models: an FVCOM4.1 Case Study," presents a straightforward solution that significantly improves the representation of hydrological influence in ocean circulation models. The authors demonstrate a keen understanding of the simplicity of this solution that would be beneficial to oceanographers when improving the results of their model. For example, the decision to utilize the grid of the atmospheric model HRDPS aligned well with this principle. While not perfect, this approach eliminates the need for remapping operations and mitigates other potential sources of error, such as water conservation.

Furthermore, the authors explore various gaps in the knowledge of the river network, including the watershed area and river mouth location. They provide valuable insights into where efforts should be concentrated to enhance the efficiency of coastal circulation models. Despite these strengths, the authors remain cognizant of the existing gaps in their rivers model.

We thank the Reviewer for their thoughtful comments and feedback.

Specific comments:

- The river proxy currently relies solely on precipitation data. Incorporating evaporation data (P-E) could enhance the proxy's accuracy in predicting the volume of water in the river system.

We agree that adding evaporation (or, as suggested by another reviewer, evapotranspiration) into the calculation would enhance the proxy's accuracy and have added a discussion of this in the manuscript. Nevertheless, we estimate that both evaporation and evapotranspiration were negligible compared to precipitation in the region during our simulation period such that either would have had a minimal role in our simulations.

The following text was added to the manuscript (new text in ***bold italics***):

Line 178ff in the revised manuscript: The proxy-calculated river discharge is illustrated in Figure 4, highlighting how much fresh water might be missing from the model inputs if only Marble River discharge was considered. ***While the proxy currently only considers precipitation, the method could benefit from using precipitation minus evapotranspiration to enhance its accuracy; however, evapotranspiration in the region was estimated to be minimal during the modelled period compared to precipitation and was not included in the current work***. ***Evapotranspiration would become more important during the spring and summer in the region***. For inputting into the model, river salinity was set to zero and river temperature was set equal to the temperature time series from the nearby Nimpkish River (Water Survey of Canada, 2023a) for all rivers and streams.

Line 378 in revised manuscript: The only requirements to estimate river and stream runoff through this approach are (1) a rudimentary knowledge of watershed area and, ideally, outpour locations, and (2) precipitation ***(or precipitation minus evapotranspiration or evaporation)*** from an atmospheric model.

- The frequency of the river proxy used in the FVCOM model is unclear. While some analyses were conducted at a high frequency (as shown in Figure 4), others were done on a monthly basis (Figure 5). Clarification on the frequency of these inputs implemented in the model would be beneficial.

We added clarification on this at lines 183 in revised manuscript (additions in ***bold italics***):

"For inputting into the model, river salinity was set to zero and river temperature was set equal to the temperature time series from the nearby Nimpkish River (Water Survey of Canada, 2023a) for all rivers and streams***. All river forcing variables (temperature, salinity and proxy-calculated river flux) had a forcing frequency of 30 minutes (e.g., see river discharge time series in Figure 4).*** "

- Insufficient time was dedicated to validating the CIOPS-W model in the region under study. This raises concerns about whether the CIOPS-W model itself could be a source of errors from the initial states and the boundary conditions used in their coastal circulation model.

We found that the Quatsino Sound model performs well at the outer mooring (QUAT1; Figure A4a in manuscript), which experiences the effects from both CIOPS-W's initial and open boundary conditions. To illustrate this point, we show comparisons of the model to temperature and salinity data from QUAT1 at two depths (40 and 100 m; Figure R2.1 below). The model also performs well at QUAT2, which is farther in the inlet and consequently only sees effects of CIOPS-W open boundary conditions (initial conditions here are based on CTD observations). We thus feel confident that CIOPS-W provides appropriate initial and boundary conditions on the shelf for the present study. That said, the current manuscript focuses on the upper 50 m of the water column inside the Sound, where the role of the open boundaries is minimal due to the typical estuarine circulation. Therefore, we do not plan to add extra material in this study to provide confidence into CIOPS-W-based initial and boundary conditions; however, we will make use of this useful comment and address this topic in our next manuscript.

[Figure]

*Figure R2.1: Comparisons of temperature and salinity from the model vs observations from QUAT1 mooring at 40 and 100 meters. This figure illustrates the model's performance over the shelf, near the mouth of the Sound, highlighting how well CIOPS-W works as initial and open boundary conditions. The time series at 40 m are also shown in Figure A4 in the manuscript.*

- The paper frequently mentions that the river proxy is rain-based and does not account for snow coverage. Conducting a climatological overview of the

region, particularly regarding winter snow coverage or snowfall, could help assess the extent of this limitation in the current model.

We added more details about the seasonal snow coverage. Bidlack et al. (2021) describes the watershed type for many watersheds along the Pacific Northwest of North America, and their results are based on 30 years of data from the Distributed Climate Water Balance Model. The data shown in their Figure 4 indicates that, averaged over 1981-2010, the Marble River and 4 smaller watersheds that drain into Quatsino Sound can be categorized as hybrid snow- and rain-driven runoff regime watersheds. This means that, on average, these 5 watersheds experience a snow melt signal in spring and elevated flow in fall/early winter from rain and rain-on-snow events. The remaining 534 watersheds in our study are rain-dominated regimes, with occasional transient snowpacks contributing minimally to stream flows. We added the following sentences briefly explaining these results from Bidlack et al. at around lines 156. This is in addition to the text we already have at lines 192-195 describing the snow coverage during the simulation period.

Lines 156ff (additions in **bold italics**): "While rain is the dominant precipitation input to the watersheds draining into ***Quatsino Sound***, ***the Marble River and four smaller watersheds (one ~150 km², the others smaller than 5 km²) are climatologically categorized as hybrid snow- and rain-driven runoff regime watersheds (based on 30 years of data, 1981-2010; Bidlack et al., 2021). These five watersheds thus experience both higher flow in fall and early winter from rain/rain-on-snow events and a snow-melt signal in spring (Bidlack et al., 2021). The remaining 534 watersheds are rain-dominated regimes with occasional transient snowpacks that contribute minimally to stream flow.*** Seasonal snowpacks ***generally*** develop above approximately 800 to 1000 m elevation, with shallow transient snowpacks at sea level every few years."

*Bidlack, A. L., Bisbing, S. M., Buma, B. J., Diefenderfer, H. L., Fellman, J. B., Floyd, W. C., Giesbrecht, I., Lally, A., Lertzman, K. P., Perakis, S. S., et al.: Climate-mediated changes to linked terrestrial and marine ecosystems across the Northeast Pacific coastal temperate rainforest margin, BioScience, 71, 581–595, https://doi.org/10.1093/biosci/biaa171, 2021.*

- Validation of the properties of deeper water should be considered. While the 0-50m depth is directly affected by river flow, this could also influence the content of deeper water through estuarine circulation.

While we agree in principle with this comment, we think that adding detailed validation of the full water column to the present manuscript will make it too long and is slightly out of scope given the focus on the top 50 m (i.e., the region where freshwater will have the largest impact).  We intend to include a more detailed, whole-column validation in an upcoming manuscript that will focus on all depths in the modelled

region. We thus added the figure and text below to the appendix as a brief evaluation of the model's performance for the whole water column.

**Appendix A5 CTD profiles: Full water column evaluation**

*Additional evaluation of modelled salinity and temperature against CTD profiles spanning the full water column showed reasonable agreement (Figure A7). The distribution of modelled salinity agrees well with the CTD observations (Figure A7a), particularly if we exclude Neroutsos Inlet from the analysis (Figure A7b). As discussed, the model shows over-mixed conditions in this deep and narrow inlet (see more in Section 5). Modelled temperature adequately compares to CTD temperature observations throughout the entire water column (Figure A7c), with a similar comparison as in the top 50 m (Figure A3a). More specifically, the model has a smaller temperature range than the observations, unable to capture the cooler (< 8ºC) and warmer (>12ºc) temperatures, in large part due to the issues encountered in Neroutsos Inlets (Figure A7d). Future work will perform a more rigorous evaluation of the full water column.*

[Figure]

*Figure A7: Histograms of (a,b) salinity (g/kg) and (c,d) temperature (ºC) for the model (green) and observations (yellow) for the entire water column. The left panels show histograms for the entire model domain whereas the right panels show histograms for the model domain excluding Neroutsos Inlet.*

- The modelling of water storage will primarily delay and temper peak precipitation, rather than limit the total volume of water flowing into the river system (line 176).

We modified this sentence (now at line187ff in the revised manuscript) accordingly to clarify this point (modifications in ***bold italics***).

"There will be some storage of the precipitation (e.g. in soil and lakes), which ***would primarily delay and*** limit the ***peak*** amount of rain that ***discharges into the fjord system***; in the particular case of the Marble River, the lakes within the watershed will partly store

some of the precipitation entering this watershed and slow the water as it flows through the system."

- A comparison with the Marble River at a higher time resolution than monthly could provide more detailed insights (line 196). One suggestion could be to combine Figures 4 and 5 for this purpose.

Rather than merging Figures 4 and 5, we added a new figure in the appendix to provide further visuals to compare the proxy to the Marble River gauge data at higher resolutions. As we mention at line 345 (lines 363 in the revised manuscript), the assumptions in this method produce at times large spikes or pulses in river discharge, most notably for the Marble River (seen in Figure 4). These spikes are unrealistic when compared to hourly discharge time series from most large rivers, which are typically much smoother. That said, the proxy is able to represent the overall magnitude of the discharge on longer timescales, for example monthly, as seen in Figure 5, and weekly, as it will be shown in the new Appendix C figure below. It is worth noting that the Marble River, which has a large watershed with three large lakes, is not a particularly good candidate for a proxy that ignores water storage. We thus do not expect the proxy to perfectly represent the Marble River discharge on shorter timescales. More complex methods (e.g., incorporating gauge data when available or employing a more sophisticated rainfall-runoff model) would be necessary to achieve an improved representation of Marble River discharge. Our analysis does, however, show that despite these limitations, the proxy delivers reasonable amounts of fresh water to the system and that the ocean model performs well with the Marble River discharge estimated from the proxy.

Additionally, while we have discussed these limitations in the Discussion (lines 355ff in revised manuscript), we now further emphasize them in the methodology section at lines 202ff in revised manuscript as follows (main changes in **_bold italics_**; editorial/style edits not highlighted here):

[revised manuscript text omitted]

- The paper states that the All Rivers configuration performs better, but then suggests that only 60-75% of the total freshwater source is necessary (line 324). An explanation of this apparent contradiction would be helpful.

We added clarification in the paragraph including line 324 (lines 334 in updated manuscript; see below). What we want to convey is that we realize that it is not always realistic/possible to include all watersheds in coastal models, but according to our tests, including at least 60-75% of total freshwater fluxes will provide similar results to a model simulation that includes all watersheds. Although these specific numbers are based on our own region, we believe that the general principle could be applied to other similar regions. Another take-home message that we want to emphasize is that the common approach of only including the major rivers (exemplified by our Marble River Only simulation) is not enough, such that efforts need to be made to add at least 60% of the incoming freshwater.

Lines 334ff (additions in **bold italics**; modifications in **bold**): Although watershed analyses established that 539 rivers and streams flow into the Quatsino Sound fjord system, **our** results indicate that **good model performance can be achieved** as long **as**

60% or more of the fresh water is accounted for. ***Compared with the Marble River Only simulation,* m**odel performance improved just by including rivers with watersheds greater than 50 km$^2$ (7 rivers total and ~58% of total flux; Figures 7-10) and including rivers with watersheds greater than 20 km$^2$ improved the model results even further (19 rivers total and ~ 75% of total flux; Figures 7-10). The latter sensitivity test performed similarly to the All Rivers simulation (Figures 6 - 10); however, it is worth noting that the Watersheds20km simulation had a larger salinity bias than the All Rivers case in the upper 50 m in both Holberg (0.5 vs. 0.2 g/kg) and Rupert (0.25 vs. <0.1 g/kg) Inlets (Figure 9). We conclude ***that including only the major rivers is not enough to achieve good coastal model performance (e.g., our Marble River Only simulation). Our results also suggest that if watershed information is limited, even including a fraction of the total freshwater sources will improve ocean model performance;*** in our case study, including at least 60-75% of total freshwater sources substantially improved ***upon the Marble River Only simulation***. While the exact amount of freshwater discharge required will depend on the specific system being modelled, these results might be useful for guiding model development for other regions.